

**Hydrologically transported dissolved organic carbon**
**influences soil respiration in a tropical rainforest**
**Running title: DOC influences soil respiration**
Author: W.-J. Zhou [1,2,3], H.-Z. Lu [1,2,3], Y.-P. Zhang [1,2*] , L.-Q. Sha [1,2*], D. Schaefer [1,2],
Q.-H. Song [1,2,3], Y. Deng [1,2], X.-B. Deng[1,2]
Affiliation:
[1.] Key Laboratory of Tropical Forest Ecology, Xishuangbanna Tropical
Botanical Garden, Chinese Academy of Sciences, Mengla, Yunnan 666303, China
[2.] Xishuangbanna Station for Tropical Rain Forest Ecosystem Studies, Chinese
Ecosystem Research Net, Mengla, Yunnan 666303, China
[3.] Graduate University of the Chinese Academy of Sciences, Beijing 100039,
China
**Corresponding author:**
Y.-P. Zhang, Tel: +86-871-65160904, Fax: +86-871-65160916, E-mail:
yipingzh@xtbg.ac.cn
L.-Q.Sha, Tel: +86-871-65160904, Fax: +86-871-65160916, E-mail:
shalq@xtbg.ac.cn
**Keywords:**
Dissolved organic carbon (DOC), Soil temperature, Soil water content, Soil
respiration, Tropical rainforest
**Paper type:**
Primary research articles



**Abstract**
To better understand the role of the dissolved organic carbon (DOC) transported by hydrological
processes in soil respiration in tropical rainforests, we measured: (1) the DOC flux in rainfall,
throughfall, litter leachate, and surface soil water (0–20 cm), (2) the seasonality of $\delta^{13}C_{DOC}$ in each
hydrological process, and $\delta^{13}C$ in leaves, litter, and surface soil, and (3) soil respiration in a
tropical rainforest in Xishuangbanna, southwest China. Results showed: The surface soil
intercepted 94.4 ± 1.2% of the annual litter leachate DOC flux and is a sink for DOC. The
throughfall and litter leachate DOC fluxes amounted to 6.81% and 7.23% of the net ecosystem
exchange, respectively, indicating that the DOC flux through hydrological processes is a key
component of the carbon budget, and may be a key link between hydrological processes and soil
respiration in a tropical rainforest. The difference in $\delta^{13}C$ between the soil, soil water (at 0–20 cm),
throughfall, and litter leachate indicated that DOC is transformed in the surface soil. The
variability in soil respiration is more dependent on the hydrologically transported DOC flux than
on the soil water content (at 0–20 cm), and is more sensitive to the soil water DOC flux (at 0–20
cm) than to the soil temperature, which suggests that soil respiration is more sensitive to the DOC
flux in hydrological processes, especially the soil water DOC flux, than to soil temperature or soil
moisture.






## 1. Introduction

Dissolved organic carbon (DOC), the most active form of fresh carbon, stimulates
microbial activity and affects $CO_2$ emissions from the surface soil (Bianchi, 2011,
Chantigny, 2003, Cleveland *et al*., 2006). This indicates that the proportion of DOC
that leaches from the soil is a crucial component of the carbon balance (Kindler *et al*.,
2011, Stephan *et al.*, 2001), which is also estimated as the high ratio of DOC flux to
net ecosystem exchange (NEE) in forests, grasslands, and croplands (Sowerby *et al.*,
2010). The DOC from water-extractable soil carbon is regenerated quickly and
functions as an important source of substrate for soil respiration (SR), especially
microbial heterotrophic respiration (HR) (Cleveland *et al*., 2004, Jandl and Sollins,
1997, Schwendenmann and Veldkamp, 2005), which contributes more to SR than
does autotrophic respiration. Laboratory studies have shown that DOC also plays a
key role in SR in the surface soil (De Troyer *et al*., 2011, Fröberg *et al*., 2005, Qiao et
al., 2013). However, most studies have been performed in the laboratory, and the
mechanisms underlying the effects of DOC on the carbon budget and SR in the field
remain unclear.
Hydrological processes that transport DOC, such as throughfall and litter leachate, are
important sources of DOC in surface soil water (De Troyer *et al.*, 2011, Kalbitz *et al*.,
2000, Kalbitz *et al*., 2007, Kindler *et al*., 2011). The soil retains most of the DOC that
reaches the soil surface from the throughfall and litter leachate (Chuyong *et al.*, 2004,





Dezzeo and Chacón, 2006, Liu and Sheu, 2003, Liu *et al.*, 2008, McJannet *et al.*, 2007,
Schrumpf *et al.*, 2006, Zimmermann *et al.*, 2007). Qiao et al. (2013) suggested that
the addition of labile organic carbon increases the decomposition of the native soil
organic carbon (SOC) by exerting a priming effect, and augments the $CO_2$ emissions
in subtropical forests. Because of the massive rainfall in tropical rainforests, more
DOC is transported to the soil by throughfall and litter leachate than in other forests.
The high temperature and leaching in tropical forests may mean that the fresh DOC
from hydrological processes affects SR differently in tropical rainforests than in
boreal and temperate forests (De Troyer *et al.*, 2011, Fröberg *et al.*, 2005, Qiao *et al.*,
2013). For this reason, research into the role of hydrologically transported DOC in the
SR in tropical rainforest is essential.
The fate of DOC intercepted by the surface soil can be determined from variations in
the DOC flux and $\delta^{13}C_{DOC}$ among soil water, soil, litter leachate, and throughfall.
Based on the seasonal and source (canopy leaf, litter, or soil) differences in $\delta^{13}C$ (De
Troyer *et al.*, 2011), $\delta^{13}C_{DOC}$ studies have shown that DOC transported from
aboveground water and from the desorption of soil aggregates is retained in the
surface soil by soil absorption or is involved in surface carbon biochemical dynamics
through soil water leaching and microbiological activity (Comstedt *et al.*, 2007, De
Troyer *et al.*, 2011, Fang *et al.*, 2009, Kindler *et al.*, 2011). This proposal has been
confirmed in a laboratory leaching experiment simulating a temperate forest, as
performed by Park *et al.* (2002), who reported that the cumulative amount of $CO_2$
evolved is positively related to the availability of carbon (Park *et al.*, 2002).

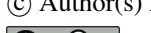



Furthermore, fresh DOC fed to the surface soil influences soil $CO_2$ emissions in both
the short term (3–14 days) and long term (month to years) (Davidson *et al*., 2012).
Therefore, several models of the surface soil carbon efflux indicate that DOC is a
factor that influences $CO_2$ emissions (Blagodatskaya *et al.,* 2011, Guenet *et al*., 2010,
Yakov, 2010) based on recent research with controlled experiments. However, the
natural mechanism underlying the effects of the hydrologically transported DOC flux
on $CO_2$ emissions remains unclear. The precipitation rate, NEE, and litterfall are all
high in tropical forests (Tan *et al*., 2010, Zhang *et al*., 2010), and several studies have
shown that DOC plays an important role in the carbon balance in these settings
(Fontaine *et al.*, 2007, McClain *et al.*, 1997, Monteith *et al.*, 2007). Here, we
investigate the relative contribution of hydrologically transported DOC to SR in a
rainforest compared with the contributions of soil temperature and moisture, which
has not been extensively studied until now.
Our study was performed in tropical rainforest at Xishuangbanna in southwest China,
on the northern edge of a tropical region. This forest has less annual rainfall (1557
mm), a smaller carbon sink (1667 kg C $ha^{-1}$) (Tan *et al*., 2010, Zhang *et al*., 2010),
lower SR (5.34 kg $CO_2$ $m^{-2}$ $yr^{-1}$) (Sha et al., 2005), and less litterfall (9.47 $\pm$ 1.65 Mg
C $ha^{-1}$ $yr^{-1}$) (Tang *et al*., 2010) than typical rainforests of the Amazon and around the
equator. We hypothesized that the ratio of throughfall and litter leachate DOC flux to
NEE is relatively high, and that hydrologically transported DOC significantly affects
SR in the tropical rainforest at Xishuangbanna. To test these hypotheses, we
determined the SR, HR, and DOC fluxes in the rainfall, throughfall, litter leachate,



and surface soil water (0–20 cm depth), the seasonal variability in $\delta^{13}$C (isotopic
abundance ratio of $^{13}$C) in DOC ($\delta^{13}C_{DOC}$) and in the carbon pools in the soil, litter,
and canopy leaves in this tropical forest.

**2 Materials and methods**

2.1 Study site

The study site is located at the center of the National Forest Reserve in Menglun,
Mengla County, Yunnan Province, China (21°56′N, 101°15′E), and has suffered
relatively little human disturbance. The weather in the study area is dominated by the
north tropical monsoon and is influenced by the southwest monsoon, with an annual
average temperature of 21.5 ℃, annual average rainfall of 1557 mm, and average
relative humidity of 86%. Based on the precipitation dynamics, the rainy season
occurs between May and October (with 84.1% of the total annual precipitation) and
the dry season between November and April.
The dominant trees are *Terminalia myriocarpa* and *Pometia tomentosa*, which are
typical tropical forest trees (Cao *et al*., 1996). The topographic slope is 12 °–18 °, and
the soil type is oxisol, formed from Cretaceous yellow sandstone, with a pH of
4.5–5.5 and a clay content (d < 0.002 mm) of 29.5% in the surface soil (0–20 cm)
(Tang *et al*., 2007).

2.2 Experimental set-up

At the study plot, three rainfall collectors were set above the canopy on a 70 m eddy



flux tower to collect rain samples. Each collector had a polytetrafluoroethylene (PTFE)
funnel (2.5 cm diameter) connected to a brown glass bottle, which was rinsed with
distilled water before each collection. To sample the throughfall, litter leachate, and
soil water (20 cm depth), four groups of replicate collectors were set for each of these
measurements. All the collectors were distributed randomly around the eddy flux
tower. The throughfall collectors were $200 \times 40$ cm$^2$ V-shaped tanks made of stainless
steel. A PTFE tube connected the collector to a polyethylene sampling barrel. The
litter leachate was collected in 40 cm $\times$ 30 cm $\times$ 2 cm PTFE plates. In the plate, we
layered 100-, 20-, and 1-mesh silica sand from the bottom to the upper edge, to a
depth of 2 cm, to ensure that the litterfall fragments did not reach the bottom of the
plate and to filter the leachate. The bottom of the plate was curved into an arc shape,
causing the leachate to flow together at the bottom funnel. The funnel was connected
by a PTFE tube to a 10 L bottle further down the slope. The soil water collector was
designed like the litter leachate collector. The collection system was buried in soil at a
depth of 20 cm along the surface slope. To reduce the disturbance from digging as
much as possible, all the soil collectors were placed in holes that were approximately
the size of the PTFE collector, and all soil was added from the bottom to the surface,
layer by layer. All the soil water and litter leachate collectors were set in place 3
months before the samples were collected, to minimize the influence of their
installation.
The water fluxes from rainfall and throughfall were estimated with an installed
water-level recorder. The recorder was set to measure the average discharge at 30 min



intervals. The daily and biweekly water fluxes from rainfall and throughfall were
calculated from the data recorded automatically between 08:00 and 08:00 on the
following day (local time). The water fluxes from the litter leachate and soil water
were determined daily by manual observation.
We set four 5m × 5 m plots around the eddy flux tower to measure SR and HR using
the trenching method. In each plot, three paired trenches and control treatments were
used to detect both HR and SR. Each treatment covered an area of $50 \times 50$ cm$^2$. Most
fine roots occur in the first 0–20 cm of soil and few occur below a depth of 50 cm in
the soil of tropical rainforests. In each trenched treatment, a polyvinyl chloride panel
was installed, and a 50-cm-deep trench was filled with *in situ* soil to protect root
respiration during the trenching treatment. Surface respiration was determined with an
Li-820 $CO_2$ Analyzer (LI–COR, Lincoln, NE, USA). From February 2008 to
February 2009, SR was detected biweekly between 10:00 and 13:00 (local time) (Sha
et al., 2005).
**Soil temperature and moisture**
Soil temperature and moisture at a depth of 5 cm were measured every 15 min with a
Campbell Scientific data logger (Campbell Scientific, North Longan, Utah, USA).
The daily average soil temperature and moisture were calculated as the daily means of
the data collected every 15 min.
**Soil, leaf, and litter sampling**
Soil (0–20 cm depth) near the soil water collectors, and leaf samples and litter
samples from around the water collector were collected in August and October, 2010,

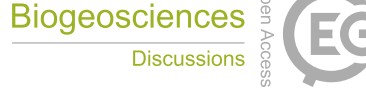

and in January, March, and May, 2011. The leaves of the dominant species were
randomly picked from the canopy around the plots, and litter samples were collected
from around the plots. Soil samples were collected with a steel foil sampler (diameter
= 5 cm, height = 20 cm). All the leaf and litter samples were oven-dried to constant
weight at 60 ℃. After drying, the leaf and litter were ground and passed through a 1
mm screen. Wind-dried soil was manually broken by hand and sieved (100 mesh) to
remove larger particles, roots, and visible soil fauna. Plant and soil samples were
analyzed for total C and $\delta^{13}$C values with an elemental analyzer (Elementar vario
PYRO cube, Germany) coupled to an continuous flow system isotope ratio mass
spectrometer (IsoPrime 100 Isotope Ratio Mass Spectrometer, Germany, EA-MS).
Samples (0.200-0.600mg dried and sieved through 100 mesh size) were wrapped in a
tin boat and loaded into the auto-sampler (EA3000, Eurovector, Milan, Italy) coupled
to the EA-IRMS. The sample was flash combusted in a combustion reactor held at
1120℃. The produced $CO_2$ was separated by the $CO_2$ absorption column, and carried
by helium to ion source for measurements. The reference $CO_2$ (>99.999%) flowed in
at 420 seconds and lasted for 30 seconds. The isotopic results are expressed in
standard notation ($\delta^{13}$C) in parts per thousand (‰) relative to the standard Pee Dee
Belemnite:
$\delta^{13}C = [^{13}R_{sample}/^{13}R_{standard} - 1] \times 1000$         (1)
where R is the molar ratio $^{13}C/^{12}C$.



**2.3 Water sampling and analysis**


All the 24 h cumulative water samples were collected at the sampling sites between
08:00 and 10:00 (local time), following the procedure outlined by Zhou et al. (2013),
using high-density polyethylene bottles. The sampling bottles were completely filled,
allowing no headspace. After the bottles were washed with 3% HCl solution, they
were rinsed with distilled water. Before sample collection, the bottles were pre-rinsed
three times with the sample water. The study was performed over three full calendar
years, from January 1, 2009, to December 31, 2011. The water samples were collected
on the day following a rain event during the dry season and once a week during the
rainy season in 2009, and once a week in 2010 and 2011. All the water samples were
immediately transported to the laboratory in insulated bags to prevent DOC
decomposition.
Based on the analytical method of Zhou *et al*. (2013), all the samples were
vacuum-filtered through a 0.45 μm glass fiber filter (Tianjinshi Dongfang Changtai
Environmental Protection Technology, Tianjin, China) and were pre-rinsed with
deionized water and the sample water under vacuum. The filtered samples were
analyzed for DOC within 24 h of collection using a total organic carbon/total nitrogen
(TOC/TN) analyzer (LiquiTOC II, Elemental Analyses System GmbH, Germany).
To analyze the water DOC isotopic $\delta^{13}$C-DOC ($\delta^{13}$C$_{DOC}$), the samples were collected
on the same day as the leaves, litter, and 0–20 cm soil samples were collected.
Subsamples (500 mL) of the rain, throughfall, litter leachate, and soil water samples
were passed through a 0.45 μm glass fiber filter and transferred to another 500 mL




polyethylene terephthalate bottle. All the filtered water was frozen and placed in a
freeze dryer until it was reduced to a fine powder. The $\delta^{13}$C of the freeze-dried DOC
was analyzed with a method similar to that for the plant and soil samples. Considering
the lower C content, more sample amount (20-60mg) were weighted, the combustion
temperature was set at 920℃, and the reference $CO_2$ flowed in at 475 seconds, laterer
than for the soil and plant samples. The sample $\delta^{13}$C abundance were calculated
according to Eq (1).
2.4 Calculations and statistics
The correlations between the following parameters were tested with Pearson's
correlation (two-tailed) and nonlinear regression tests: the daily water flux and DOC
concentration, SR, HR, soil moisture, and soil temperature from February 2008 to
January 2009, and the biweekly SR and HR rates and the amounts of DOC and water
in 2009–2011. One-way analysis of variance (ANOVA) was used to compare the
hydrological DOC fluxes and $\delta^{13}$C$_{DOC}$ among different hydrological processes. The
dry season and rainy season data were compared with a paired $t$ test. The SPSS 15.0
software was used for all calculations.
Because the individual correlations between the water flux and the DOC
concentration in the throughfall, litter leachate, and soil water were significant (Fig.
S1), the regression equations used for the water flux and DOC concentration (Y = ae$^{bx}$)
were as follows:
$C_{TF} = 48.69e^{-0.097x}$     adjusted $r^2 = 0.3883$, $p = 0.002$                    (2)



$C_{LL} = 60.93e^{-0.048x}$     adjusted $r^2 = 0.4131$, $p < 0.001$        (3)
$C_{sw} = 6.78e^{-0.02048x}$     adjusted $r^2 = 0.5840$, $p < 0.001$        (4)
where $C_{TF}$, $C_{LL}$, and $C_{sw}$ are the DOC concentrations (mg L$^{-1}$) in the throughfall, litter
leachate, and soil water (0–20 cm), respectively, and x is the water flux per day (mm).
We did not collect all the individual rainfall events, throughfall, litter leachate, and
soil water samples to analyze the DOC concentrations, but interpolated all the DOC
concentrations and water fluxes according to eq(2)–(4).
The daily DOC flux was calculated as
$F = CV/100$                  (5)
where F is the daily DOC flux (kg C ha$^{-1}$ d$^{-1}$), C is the DOC concentration (mg L$^{-1}$),
and V is the water flux (mm d$^{-1}$) per day.
The biweekly carbon flux was calculated as the sum of the daily DOC fluxes.
The correlations between soil temperature at a depth of 5 cm and both SR and HR
were strong (Fig. S2) between February 2008 and January 2009. SR and HR during
the period from January 1, 2009 to December 31, 2011 were calculated based on the
equation $Y = ae^{bx}$ from the data collected between February 2008 and January 2009,
as follows:
$SR = 46.37e^{(0.11T5)}$ $r^2 = 0.8966$, $p < 0.0001$        (6)
$HR = 18.90e^{(0.14T5)}$ $r^2 = 0.8372$, $p < 0.0001$        (7)
where SR is total soil respiration (mg CO$_2$ m$^{-2}$ s$^{-1}$), HR is heterotrophic respiration
(mg CO$_2$ m$^{-2}$ s$^{-1}$), and T5 is soil temperature at 5 cm depth.




**3 Results**
3.1 Water and DOC fluxes in a tropical rainforest
The seasonal and annual water fluxes decreased from the rainfall to the surface soil
(Fig. 1a). The interception rate of the water between hydrological processes was
higher in the dry season than in the rainy season (Fig. 1a). The highest interception
rate was between the litter leachate and the surface soil ($63.85 \pm 7.98\%$), which was
$62.19 \pm 15.07\%$ in the rainy season and $81.64 \pm 23.38\%$ in the dry season.
The seasonal dynamics of the DOC flux were similar to those of the water flux (Fig.
1). The annual DOC flux increased from rainfall ($41.9 \pm 3.8$kg C ha$^{-1}$ yr$^{-1}$) to
throughfall ($113.5 \pm 8.5$ kg C ha$^{-1}$ yr$^{-1}$) and to litter leachate ($127.7 \pm 8.5$ kg C ha$^{-1}$
yr$^{-1}$), and then decreased sharply to the surface soil at 0–20 cm ($7.07 \pm 1.4$ kg C ha$^{-1}$
yr$^{-1}$) (Fig. 1b). The surface soil intercepted most of the DOC coming from the
previous layer (annual: $94.4 \pm 1.2\%$, dry season: $96.7 \pm 4.4\%$, rainy season: $93.9 \pm$
2.6%). That the interception rates for water and DOC were greatest in the surface soil
indicates that the surface soil is the most important water and DOC sink in this
tropical rainforest.
3.2 Isotopic characteristics of DOC in the hydrological processes of a tropical
rainforest
During the transfer of rainfall to soil water (0–20 cm), $\delta^{13}C_{DOC}$ was highest in the
rainfall DOC and lowest in the throughfall DOC in both the rainy and dry seasons





(Table 1). The seasonal difference in $\delta^{13}C_{DOC}$ was highest in the surface soil water
(3.25‰) and lowest in the litter leachate (0.11‰). From the litter leachate to the
surface soil water, $\delta^{13}C_{DOC}$ increased significantly by 4.26‰ ($p = 0.05$) in the rainy
season, but increased by only 1.12‰ (not significant, $p = 0.39$) in the dry season. $\delta^{13}C$
increased from the canopy leaves to the soil and did not differ significantly between
seasons (Table 1).
In both the dry and rainy seasons, $\delta^{13}C_{DOC}$ in water was higher than $\delta^{13}C$ in the
corresponding element (comparing throughfall with leaves, litter leachate with litter,
and soil water with soil at 20 cm depth) (Table 1). The smallest difference between
$\delta^{13}C_{DOC}$ and $\delta^{13}C$ in each compartment occurred between soil water DOC and soil
carbon in the dry season, which was only 0.23‰. The greater difference between
$\delta^{13}C_{DOC}$ and $\delta^{13}C$ in the rainy season than in the dry season for soil water and soil
(Table 1) indicates that the biogeochemical dynamics of DOC are more active in the
rainy season than in the dry season in soil.
3.3 Surface soil $CO_2$ flux dynamics in a tropical rainforest
In the tropical rainforest at Xishuangbanna, SR was dominated by HR (Fig. 2). HR
contributed more to SR during the rainy season (76.8 ± 0.8%) than during the dry
season (66.5 ± 0.5%), and the annual contribution of HR to SR was 71.7 ± 0.7%. This
indicates that HR is more important to the surface $CO_2$ flux than is root respiration.
SR and HR were higher in the rainy season than in the dry season, similar to the
dynamics of the hydrological and DOC fluxes (Fig. 1).



Soil temperature explained 89.0% and 84.0% of the variation in SR and HR,
respectively, and soil moisture explained 37.6% and 31.2% of the variation in SR and
HR, respectively (Table 2). The sensitivity indices of SR and HR for soil temperature
at a depth of 10 cm were 3.00 and 4.06, respectively, whereas their sensitivity indices
for soil moisture were 1.32 and 1.37, respectively, based on observational data (Table
2, Fig. S2).
3.4 Influence of DOC flux on soil $CO_2$ flux in a tropical rainforest
There were significant correlations between the mean biweekly SR and HR and the
biweekly water fluxes and DOC fluxes through the hydrological processes (Table 2).
Based on the definition of the t,emperature-dependent sensitivity index ($Q_{10}$) for SR,
which is the increase in SR caused by a 10 ℃ increase in temperature, we also
defined a soil-water-content-dependent sensitivity index, a DOC-flux-dependent
sensitivity index, and a water-flux-dependent sensitivity index in this study, analogous
to the temperature-dependent sensitivity index for SR (Table 2). An independent $t$ test
showed that the DOC-flux-dependent sensitivity indices for SR (3.62 ± 4.36) and HR
(5.12 ± 7.12) were significantly higher than the water-flux-dependent sensitivity
indices for SR (1.07 ± 0.019, F = 8.91, $p$ = 0.024) and HR (1.09 ± 0.022, F = 8.94, $p$ =
0.024), respectively, which indicates that SR and HR were more sensitive to the DOC
flux than to the water flux through the hydrological processes. No significant
difference was observed between the water-flux-dependent indices for SR (1.07 ±
0.019) and HR (1.09 ± 0.022), or between the DOC-flux-dependent indices for SR



(3.62 ±4.36) and HR (5.12 ±7.12).
The soil-water-content-dependent sensitivity indices for HR and SR were higher than
all the water-flux-dependent sensitivity indices, and smaller than the
DOC-flux-dependent sensitivity indices for SR and HR (Table 2). This indicates that
SR and HR are more sensitive to the DOC flux than to the soil water content. The
soil-temperature-dependent sensitivity indices for HR and SR were higher than all the
water-flux- and DOC-flux-dependent sensitively indices, except the soil-water (0–20
cm)-DOC-flux-dependent sensitivity index. A comparison of the sensitivity indices
for water flux, DOC flux, soil temperature, and soil moisture in all the hydrological
processes reveals that SR and HR were most sensitive to the DOC flux dynamics in
the soil water (0–20 cm depth) when biweekly variations in the Xishuangbanna
tropical rainforest were considered.
**4 Discussion**
Our results showed that the throughfall carried most of the DOC (113.5 ± 8.5 kg C
ha$^{-1}$ yr$^{-1}$) through the hydrological processes in the Xishuangbanna tropical rainforest,
which amounted to 6.81% of the NEE ($1.67 \times 10^3$ kg C ha$^{-1}$ yr$^{-1}$) (Tan e*t al*., 2010) in
this tropical rainforest in southwest China. The litter leachate DOC (127.7 ±8.5 kg)
accounted for 7.23% of the NEE in this forest. This result indicates that the
throughfall DOC is a key component of the tropical rainforest carbon budget. The
litter leachate fed a great deal of DOC to the soil, but the surface soil intercepted 94.4
± 1.2% (127.7 ±8.0 kg) of the DOC, and the surface soil water DOC flux was only



7.1±1.4 kg C ha$^{-1}$ yr$^{-1}$, which was slightly less than that at the headwater stream outlet
(10.31 kg C ha$^{-1}$ yr$^{-1}$) (Zhou *et al*., 2013). The surface soil intercepted the bulk of the
litter leachate DOC and transported little DOC to the deep layer, indicating that the
surface soil is the DOC sink in the tropical rainforest in Xishuangbanna.
The small seasonal differences in $\delta^{13}C_{DOC}$ in the rainfall, throughfall, and litter
leachate indicate that the DOC in the aboveground water is seasonally stable (Table 1).
However, $\delta^{13}C_{DOC}$ in the soil water (at 0–20 cm) was higher in the rainy season
(3.25‰) than in the dry season, indicating that the DOC reaction in the surface soil is
seasonal. In the dry season, $\delta^{13}C_{DOC}$ in the surface soil water (–27.1 ± 2.2‰) was
similar to $\delta^{13}C_{DOC}$ in the soil (–27.3 ± 0.1‰), indicating that the soil is the major
source of soil water DOC. This is attributable to the combined absorption effects of
the high clay content (Fröberg, 2004, Lemma *et al*., 2007, Sanderman and Amundson,
2008, Tang *et al*., 2007) and the lack of water carrying DOC through the different
compartments in the dry season. Therefore, most DOC is locally produced rather than
transported. Less water and the lower DOC input from litter leachate and throughfall
to the surface soil (Fig. 1) also contribute to a reduction in microbial activity, which
contributes negligibly to the soil DOC when the soil moisture and soil temperature are
low in the dry season (Wu *et al.*, 2009). In the rainy season, the soil water content and
soil temperature are higher, so there is more vigorous biogeochemical activity in the
surface soil (Bengtson and Bengtsson, 2007). Therefore, more DOC is released from
the soil to be mineralized by microorganisms, and there is more $^{13}$C in the soil water
DOC ($\delta^{13}$C = –23.9 ± 2.2‰) than in the soil ($\delta^{13}$C, –27.3 ± 0.1‰). The relatively low



$\delta^{13}C_{DOC}$ in the litter leachate ($\delta^{13}C_{DOC}$ = −28.1 ± 2.7‰) compared with the soil water
indicates that the DOC from the litter leachate has attended in the carbon cycle in the
surface soil (Cleveland *et al.*, 2006, De Troyer *et al.*, 2011). Furthermore, most of the
DOC from the throughfall, litter leachate, and litter was fed to the surface soil, and the
soil water $\delta^{13}C_{DOC}$ value was higher than that of the throughfall, litter leachate DOC,
and $\delta^{13}C$ soil (0–20 cm) values (Table 1). These data indicate that all the DOC
transported by the throughfall and litter leachate was ultimately involved in the
surface soil carbon cycle (Fröberg *et al.*, 2003, 2005, Kammer *et al.*, 2012), and has
also contributed to the SR because it is an important part of the surface soil carbon
cycle in the tropical rainforest at Xishuangbanna.
Laboratory-based studies of tropical forests have shown that DOC primes the soil $CO_2$
flux (Qiao *et al.*, 2013). A study of a temperate forest showed that the rate of DOC
production is one of the rate-limiting steps for SR (Bengtson and Bengtsson, 2007).
Comparative studies of $^{13}C$ and $^{14}C$ in DOC and SOC have also shown that fresh
organic carbon stimulates the activity of old carbon, and increases the emission of
$CO_2$ because DOC is the substrate of microbial activity (Cleveland *et al.*,2004, 2006,
Hagedorn and Machwitz, 2007, Hagedorn *et al.*, 2004, Qiao *et al.*, 2013). Because the
microbial biomass and potential carbon mineralization rates are higher in soils with
higher DOC contents than in soils with lower DOC contents (Montaño *et al.*, 2007),
the DOC turnover rate (Bengtson and Bengtsson, 2007) is rapid and the
transformation period is short (3–14 days) (Cleveland *et al.*, 2006, De Troyer *et al.*,
2011). This indicates that DOC is involved in the surface soil carbon cycle in the short





term by affecting SR (Cleveland *et al.*, 2004, 2006). Although we did not determine
the period of the DOC turnover cycle, the biweekly DOC flux passing through the
hydrological processes (throughfall, litter leachate, soil water, and interception by the
surface soil) significantly explained SR and HR, with higher sensitively indices than
the indices for the soil water content and water flux (Table 2), predicting that DOC
has a significant impact on soil $CO_2$ emissions in this tropical rainforest.
The DOC-flux-dependent sensitivity indices for the different parts of the hydrological
processes in this tropical rainforest were higher than the amount-of-water-dependent
sensitivity indices, which shows that the DOC flux affects SR more significantly than
the amount of water passing through the system, because of the combined effects of
water and DOC on SR.
It is important to consider which part of the DOC flux in the hydrological processes of
this tropical rainforest most strongly influences SR. Previous studies have shown that
of all the factors affecting SR, it is most sensitive to soil temperature (Bekku *et al.*,
2003, Reichstein *et al*., 2003, Zheng *et al.*, 2009), as in the tropical forest at
Xishuangbanna (Sha *et al.*, 2005). Although soil temperature better explained SR and
HR than the DOC flux, the sensitivity indices for the soil water DOC fluxes were
higher than the sensitivity indices for soil temperature, although temperature
explained the rate of SR better than the DOC flux (Table 2). At this study site, HR,
which depends predominantly on microbial activity and substrates, contributed the
major fraction of SR (Fig. S2), so not only HR, but also SR depends most strongly on
the microbial and respiratory substrates in this tropical rainforest. Therefore, the DOC





transported by the forest hydrological processes, from litter decomposition, root
exudates, and the soil itself, will contribute to SR (Table 1). The bioavailability of the
DOC transported by hydrological processes is greater than that of SOC (De Troyer *et*
*al.*, 2011, Kindler *et al.*, 2011). The DOC from throughfall and litter leachate is also
an important contributor because $\delta^{13}C_{DOC}$ differs between the surface soil water and
the litter leachate and throughfall (Table 1). Although ectotrophic mycorrhizae
contribute significantly to SR in the rhizospheres of some temperate and boreal forests
(Neumann *et al.*, 2014; Tomè *et al.*, 2016), in this tropical rainforest, EMF:
*Paraglomus,* a kind of endomycorrhiza, occupies more than 90% of the mycorrhizal
community (Shi, 2014). Together with roots, and root exudate, it contributes to the
autotrophic SR, which is only 28.9% of the total SR, so the mycorrhiza is not the
dominant contributor to SR in this tropical rainforest. The other details of the
biogeochemical processes affecting DOC in the surface soil are not obvious in this
study. However, according to both laboratory and field studies, the DOC intercepted
by the surface soil clearly affects HR (Table 2), together with the DOC from litter
decomposition and the soil itself (Cleveland *et al.*, 2004, 2006, Hagedorn and
Machwitz, 2007, Hagedorn *et al.*, 2004, Jandl and Sollins, 1997, Keiluweit *et al.*,
2015; Montaño *et al.*, 2007, Qiao *et al.*, 2013, Schwendenmann and Veldkamp, 2005;).
Considering the effect of DOC on SR, the surface soil water DOC is the most
sensitive index of HR and SR (Table 2). The details of the biogeochemical processes
affecting DOC in the surface soil are not obvious in this study. However, according to
both laboratory and field studies, the DOC intercepted by the surface soil clearly



affects SR (Table 2), together with the DOC from litter decomposition and the soil
itself (Cleveland *et al.*, 2004, 2006, Hagedorn and Machwitz, 2007, Hagedorn *et al.*,
2004, Jandl and Sollins, 1997, Montaño *et al.*, 2007, Qiao *et al.*, 2013,
Schwendenmann and Veldkamp, 2005). Considering the effect of DOC on SR, the
surface soil water DOC is the most sensitive index of HR and SR (Table 2).
This study demonstrates that the surface soil is a sink for the DOC transported by
hydrological processes (Fig. 1), and that HR and SR are sensitive to the DOC flux
through these processes. The most sensitive indicator of SR is the soil water DOC
flux (at 0–20 cm), exceeding the sensitivity of the soil temperature, soil water content,
water flux, and DOC flux along all the hydrological processes (Table 2). The
variations in $\delta^{13}$C in DOC, soil, and plants also partly support the notion that the soil
water DOC flux is the most sensitive index of SR in this tropical rainforest. The
results suggest that the DOC transported by hydrological processes plays the most
important role in the SR processes. In the context of global climate change, more
attention must be paid to the contribution of hydrologically transported DOC in future
studies of the mechanisms of SR.
**Author contribution**
W.-J. Zhou and D. Schaefer, H.-Z. Lu, Sha L-Q, Y.-P Zhang designed the
experiments and W.-J. Zhou, H.-Z. Lu, Q.-H. Song, Y. Deng, X.-B. Deng carried
them out. W-J Zhou prepared the manuscript with contributions from all co-authors.



**Acknowledgments**
This work was supported by the National Natural Science Foundation of China
(41271056, U1202234, 40801035), the Strategic Priority Research Program of the
Chinese Academy of Sciences (No. XDA05020302 and XDA05050601), the Natural
Science Foundation of Yunnan Province (2015FB188), the CAS 135 Project
(XTBG-F01), and the Science and Technology Service Network Initiative of the
Chinese Academy of Sciences (No. KFJ-EW-STS-084).
We thank the staff and technicians of the Xishuangbanna Station for Tropical Rain
Forest Ecosystems who assisted with field measurements and the Public Technology
Service Center of Xishuangbanna Tropical Botanical Garden, CAS, who contributed
to [13]C analyses. We also thank Zhi-Gang Chen and Zhi-Hua Zhou for assistance with
sampling, Zhi-Ling Chen and Li-Fang OU for laboratory work, and Jan Mulder and
Jing Zhu for reviewing the manuscript.

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

Lemma, B., Nilsson, I., Kleja, D.B., Olsson, M., Knicker, H.: Decomposition and
substrate quality of leaf litters and fine roots from three exotic plantations and a
native forest in the southwestern highlands of Ethiopia, Soil Bio. Bioch., 39,
2317–2328, 2007.
Liu, C.P., Sheu, B.H., 2003. Dissolved organic carbon in precipitation, throughfall,
stemflow, soil solution, and stream water at the Guandaushi subtropical forest in
Taiwan, Forest Eco. Manag., 172, 315–325, 2003.
Liu, W.J., Liu, W.Y., Li, J.T., Wu, Z.W., Li, H.M.: 2008. Isotope variations of
throughfall, stemflow and soil water in a tropical rain forest and a rubber





plantation in Xishuangbanna, SW China, Hydrol. Res., 39, 437–449, 2008.
McClain, M.E., Richey, J.E., Brandes, J.A., Pimentel, T.P.: Dissolved organic matter

and terrestrial-lotic linkages in the central Amazon basin of Brazil, Global

Biogeochem. Cy., 11, 295–311, 1997.

McJannet, D., Wallace, J., Reddell, P.: Precipitation interception in Australian tropical

rainforests: I. Measurement of stemflow, throughfall and cloud interception,

Hydrol. Process., 21, 1692–1702, 2007.

Montaño, N.M., García-Oliva, F., Jaramillo, V.J.: Dissolved organic carbon affects

soil microbial activity and nitrogen dynamics in a Mexican tropical deciduous

forest, Plant Soil, 295, 265–277, 2007.

Monteith, D.T., Stoddard, J.L., Evans, C.D., de Wit, H.A., Forsius, M., Høgåsen, T.,

Wilander, A., Skjelkvåle, B.L., Jeffries, D.S., Vuorenmaa, J., Keller, B., Kopácek,

567        J., Vesely, J.: Dissolved organic carbon trends resulting from changes in

atmospheric deposition chemistry, Nature, 450, 537–540. 2007.

Neumann J., Matzner E.: Contribution of newly grown extramatrical ectomycorrhizal

mycelium and fine roots to soil respiration in a young Norway spruce site, Plant

soil, 378(1–2): 73–82, 2014.

Park, J.H., Kalbitz, K., Matzner, E.: Resource control on the production of dissolved

organic carbon and nitrogen in a deciduous forest floor, Soil Bio. Bioche., 34,

1391–1391, 2002..

Qiao, N., Schaefer, D., Blagodatskaya, E., Zou, X., Xu, X., Kuzyakov, Y.: Labile

carbon retention compensates for $CO_2$ released by priming in forest soils, Global



Change Biol., 20(6): 1943–1954, 2014..
Reichstein, M., Rey, A., Freibauer, A., Tenhunen, J., Valentini, R., Banza, J., Casals, P.,

Cheng, Y.F., Grunzweig, J.M., Irvine, J., Joffre, R., Law, B.E., Loustau, D.,

Miglietta, F., Oechel, W., Ourcival, J.M., Pereira, J.S., Peressotti, A., Ponti, F., Qi,

Y., Rambal, S., Raymant, M., Romanya, J., Rossi, F., Tedeschi, V., Tirone, G., Xu,

582         M., Yakir, D.: Modeling temporal and large-scale spatial variability of soil

respiration from soil water availability, temperature and vegetation productivity

indices, Global Biogeoche. Cy., 17, DOI: 10.1029/2003GB002035,2003.

Sanderman, J., Amundson, R.: A comparative study of dissolved organic carbon

transport and stabilization in California forest and grassland soils,

Biogeochemistry, 89, 309–327, 2008.

Schrumpf, M., Zech, W., Lehmann, J., Lyaruu, H.V.C.: TOC, TON, TOS and TOP in

rainfall, throughfall, litter percolate and soil solution of a montane rainforest

succession at Mt. Kilimanjaro, Tanzania, Biogeochemistry, 78, 361–387, 2006.

Schwendenmann, L., Veldkamp, E.: The role of dissolved organic carbon, dissolved

organic nitrogen, and dissolved inorganic nitrogen in a tropical wet forest

ecosystem, Ecosystems, 8, 339–351, 2005..

Sha, L.Q., Zheng, Z., Tang, J.W., Wang, Y.H., Zhang, Y.P., Cao, M., Wang, R., Liu,

G.G., Wang, Y.S., Sun, Y.: Soil respiration in tropical seasonal rain forest in

Xishuangbanna, SW China, Science in China Series D-Earth Sciences, 48,

189–197, 2005(In Chinese).

Shi L.L.: Soil Microbial Community in Forest Ecosystem as Revealed by Molecular



Techniques —diversity pattern, maintenance mechanism, and the response to
disturbance, A Dissertation Submitted to University of Chinese Academy of
Sciences, In partial fulfillment of the requirement Doctor of Philosophy, 2014 (In
Chinese).
Sowerby, A., Emmett, B.A., Williams, D., Beier, C., Evans, C.D.: The response of
dissolved organic carbon (DOC) and the ecosystem carbon balance to
experimental drought in a temperate shrubland, Eur. J. Soil Sci., 61, 697–709,

2010.

Stephan, S., Karsten, K., Egbert, M.: Controls on the dynamics of dissolved organic
carbon and nitrogen in a Central European deciduous forest, Biogeochemistry, 55,
327–349, 2001.
Tan, Z., Zhang, Y., Yu, G., Sha, L., Tang, J., Deng, X., Song, Q.: Carbon balance of a
primary tropical seasonal rain forest, J. Geophys. Res, 115 (D4),
DOI: 10.1029/2009JD012913, 2010.
Tang, J.W., Cao, M., Zhang, J.H., Li, M.H., Litterfall production, decomposition and
nutrient use efficiency varies with tropical forest types in Xishuangbanna, SW
China: a 10-year study, Plant soil, 335, 271–288, 2010.
Tang, Y.L., Deng, X.B., Li, Y.W., Zhang, S.B.: Research on the difference of soil
fertility in the different forest types in Xishuangbanna, Journal of Anhui
Agricultural Sciences, 35, 779–781,2007 (In Chinese).
Tomè, E., Ventura, M., Folegot, S., Zanotelli, D., Montagnani, L., Mimmo, T., Tonon,
G., Tagliavini, M., Scandellari, F.: Mycorrhizal contribution to soil respiration in




an apple orchard. Appl. Soil. ecol.,, 101, 165–173, 2016.
Wu, Y., Yang, X., Yu, G.: Seasonal fluctuation of soil microbial biomass carbon and its
influence factors in two types of tropical rainforests, Ecology and Environmental
Sciences, 18, 658–663, 2009 (In Chinese).
Yakov, K.: Priming effects: Interactions between living and dead organic matter. Soil
Bio. Biochem. 42, 1363–1371, 2010.
Zhang, Y., Tan, Z., Song, Q., Yu, G., Sun, X.: Respiration controls the unexpected
seasonal pattern of carbon flux in an Asian tropical rain forest, Atmos. Environ.
44, 3886–3893, 2010.
Zheng, Z.M., Yu, G.R., Fu, Y.L., Wang, Y.S., Sun, X.M., Wang, Y.H.: Temperature
sensitivity of soil respiration is affected by prevailing climatic conditions and
soil organic carbon content: A trans-China based case study, Soil Bio. Biochem.,
41, 1531–1540, 2009.
Zimmermann, A., Wilcke, W., Elsenbeer, H.: Spatial and temporal patterns of
throughfall quantity and quality in a tropical montane forest in Ecuador, J.
Hydrol. 343, 80–96, 2007.



**Table legends**
**Table 1** DOC $\delta^{13}$C dynamics along the hydrological processes (R, rainfall, TF, throughfall, LL, litter
leachate) and the $\delta^{13}$C in leaves, litter, and surface soil in the tropical rainforest at Xishuangbanna,
southwest China
**Table 2** Results of a regression analysis of the biweekly water flux, DOC flux, soil respiration (SR),
and heterotrophic respiration (HR) along the hydrological processes (TF, throughfall, LL, litter leachate)
in the tropical rainforest in Xishuangbanna, southwest China.
**Figure captions**
**Figure 1** Amount of water (A) and DOC flux along the hydrological processes in the tropical rainforest
at Xishuangbanna, southwest China.
**Figure 2** Dynamics of soil respiration (SR) and heterotrophic respiration (HR) in the tropical rainforest
at Xishuangbanna, southwest China.
The shaded area indicates the rainy season.



Table 1

| Season | R | TF | LL | Soil water (0–20 cm) | Leaves | Litter | Soil (0–20 cm) |
|---|---|---|---|---|---|---|---|
| Rainy season | −23.9±3.3 | −28.7±1.7 | −28.1±2.7 | −23.9±1.6 | −32.4±0.6 | −30.4±0.2 | −27.3±0.1 |
| Dry season | −23.8±1.3 | −29.1±1.6 | −28.1±1.5 | −27.1±2.2 | −32.5±0.5 | −30.2±0.1 | −27.3±0.1 |

R indicates rainfall, TF indicates throughfall, LL indicates litter leachate, SW20 indicates soil water at
a depth of 20 cm.






Table 2

| | Parameters | | a | b | $r^2$ | *P* | Sensitivity index |
|---|---|---|---|---|---|---|---|
| | TF | SR | 354.80 | 0.0064 | 0.4271 | <0.0001 | 1.07 |
| | | HR | 252.17 | 0.0075 | 0.4178 | <0.0001 | 1.08 |
| | LL | SR | 380.19 | 0.0058 | 0.2469 | <0.0001 | 1.06 |
| Water | | HR | 273.98 | 0.0068 | 0.2556 | <0.0001 | 1.07 |
| flux | LL-SW20 | SR | 375.65 | 0.0095 | 0.2525 | <0.0001 | 1.10 |
| | | HR | 269.67 | 0.0112 | 0.2469 | <0.0001 | 1.12 |
| | SW20 | SR | 404.49 | 0.0062 | 0.1194 | <0.0001 | 1.06 |
| | | HR | 294.92 | 0.0073 | 0.119 | <0.0001 | 1.08 |
| | TF | SR | 341.71 | 0.0441 | 0.4681 | <0.0001 | 1.55 |
| | | HR | 240.71 | 0.0524 | 0.459 | <0.0001 | 1.69 |
| | LL | SR | 355.26 | 0.0316 | 0.4061 | <0.0001 | 1.37 |
| | | HR | 251.47 | 0.0405 | 0.3971 | <0.0001 | 1.50 |
| | LL-SW20cm | SR | 355.00 | 0.0336 | 0.4021 | <0.0001 | 1.40 |
| | | HR | 251.47 | 0.0402 | 0.3971 | <0.0001 | 1.49 |
| DOC flux | SW20 | SR | 392.50 | 0.2318 | 0.2053 | <0.0001 | 10.16 |
| | | HR | 284.09 | 0.276 | 0.276 | <0.0001 | 15.80 |
| | T5 | SR | 46.37 | 0.11 | 0.89 | <0.0001 | 3.00 |
| | | HR | 18.90 | 0.14 | 0.84 | <0.0001 | 4.06 |
| | swc | SR | 153.30 | 0.0274 | 0.3756 | 0.0004 | 1.32 |
| | | HR | 96.85 | 0.0314 | 0.3199 | 0.0013 | 1.37 |

Equations used to calculate the sensitivity indices: sensitivity index = $e^{(10b)}$, where b is the parameter of
the regression equation for SR, and for soil temperature, soil water content, water flux, and DOC flux:
$Y = ae^{bx}$, where Y is the soil respiration rate, and x is soil temperature, soil water content, water flux, or
DOC flux.
TF indicates throughfall, LL indicates litter leachate, SW20 indicates soil water at a depth of 20 cm,
LL- SW20cm indicates the difference between litter leachate and soil water at a depth of 20 cm, T5
indicates soil temperature at a depth of 5 cm, SWC indicates soil water content.





Figure 1

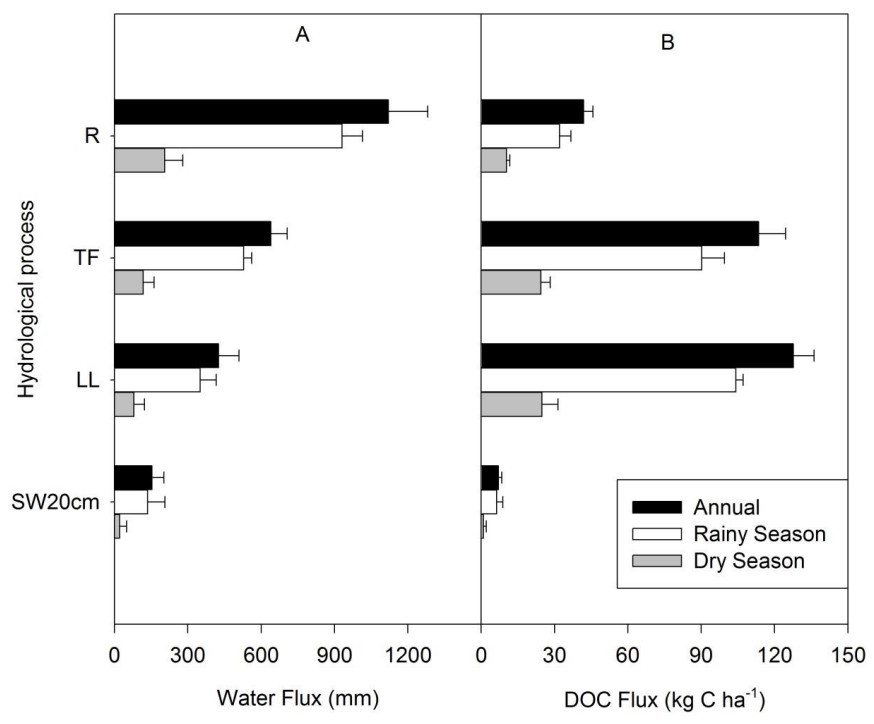







670        Figure 2

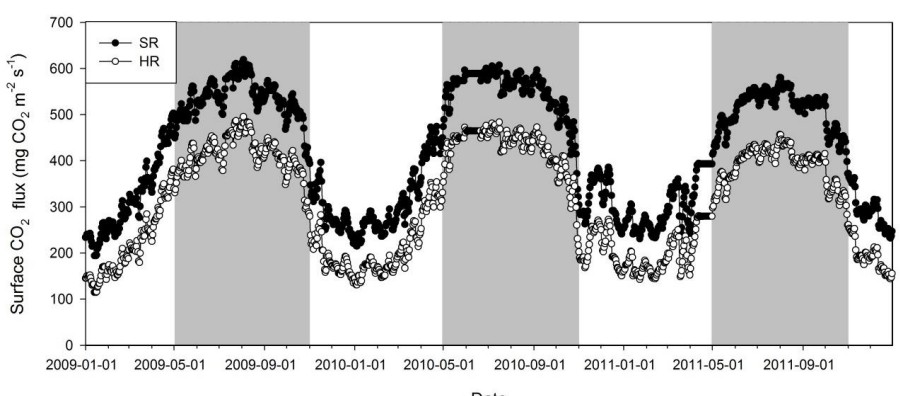

