# Peer review of "Hydrologically transported dissolved organic carbon influences soil respiration in a tropical"

_Biogeosciences, 2016_

## Referee Comment (RC1) · Anonymous Referee #1 · 8 Jul 2016

This manuscript into the effects of soil organic carbon on soil respiration represents an interesting work, and overall, it was nice to read. It showed some interesting results, in that the authors found out, that in their tropical sites, DOC fluxes in the soil water was one the most important factors explaining the variability in soil respiration, when looking at calculated sensitivity indices. Nonetheless, also soil temperature and soil moisture were of great importance, when looking at the regressions themselves.

However, there were several issues, affecting on the quality of this manuscript: The most important issue was the statistical testing: the use of one-way Anova seems to be not really appropriate for this kind of time-series data. I would suggest using a linear mixed model with repeated measures. Missing or unbalanced data are usually

no problem for this kind of analyses. The second point is that, although the authors made some statistical testing, it hardly was shown anywhere. Please show the results, either in a table or incorporated into the text. The third issue was that I was missing data on soil temperature and moisture. For example, it could be easily incorporated into Figure 2, as such. Finally, it was not clear to me who the authors calculated all the sensitivity indices.

More detailed comments: lines 24–28: this sentence is not easy to understand for the reader line 24: "role" could be changed to "effect" line 25: "in" could be changed to "on" line 27: what processes do you mean? line 28: what do you mean by "surface soil"? lines 52 and 54: first you state that laboratory studies have shown, later you write: however, most studies have been performed in the laboratory... line 66: do you mean both in terms of absolute and relative numbers? line 118: do you have any additional tree data, like age or tree density? line 137: how ling were the tubes? line 156: you removed the roots? line 198: how often they occurred during the dry season? line 221: how this was calculated? Weekly divided by 7? lines 221-224:from this sentence it is not entirely clear, what was compared to what line 259: is this annual average? lines 256-269: how about putting interception values into a table for better comparison? lines 272–287: somehow I could not follow all these differences from Table 1 line 292: this is already discussion, please move it there lines 304-308: it was not clear to me how you calculated the sensitivity indices Tables and figures: please add results of statistical tests Fig.S1: what about a possible dilution effect, resulting in lower DOC with more water?

---

## Referee Comment (RC2) · Anonymous Referee #1 · 8 Jul 2016

This comment is about Figure S2: why there is a gap in the temperature data? No data is presented between 18 and 21 degrees?

---

## Referee Comment (RC3) · Anonymous Referee #2 · 8 Jul 2016

Zhou et al. quantified dissolved organic carbon (DOC) fluxes in precipitation, through-fall, litter leachate and soil water, and linked them to soil respiration for a tropical rain forest in southwest China. In addition, they also measured 13C natural abundance for DOC and for plant and soil to examine the sources of DOC. This study is within the scopes of the journal, and interesting to the audiences who are working on tropical forest C cycling. This manuscript was well organized and written. However, I have two major concerns. One is about the sensitivity index. We know that soil respiration increases with increasing temperature, and Q10 is widely used to determine the temperature sensitivity. The authors developed similar sensitivity index for soil respiration to water fluxes, DOC fluxes and soil water content. I believe that these kinds

of sensitivity index are useful when comparing them among different sites, as is the Q10. However, I don't think we can compare among the temperature sensitivity, soil water content sensitivity, water flux sensitivity, DOC flux sensitivity within the same site, because they are different parameters and the units for each parameter are different. Thus the authors need to provide the rationales for these comparisons, otherwise the conclusions stated by lines 35 to 38 are different to stand. The other concern is about the importance of DOC. DON input from throughfall accounted for about 7% of the net ecosystem C exchange. However, it may be even minor when compared to soil respiration. So it needs to not overstate the importance of DOC in C budget. The phrase of "key" in the abstract (line 32) and throughout the manuscript may be not proper, to my point of view. It may be better to use "important" instead "key".

Specific comments: 1) Line 96, in a tropical forest; 2) Line 124, how large is your study plot? 3) Line 127, "the" may be not necessary; 4) Line 179, 2 to 6 mg? what standards were used to calibrate the measured values for plant and soil samples, as well as for DOC samples? 5) Line 291, the contribution of HR to total soil respiration was 72%, which is in higher than many reports for forests? Is this normal? 6) Line 299, sensitivity of soil respiration to soil moisture has not shown in Fig S2. 7) Line 309-310, how did you calculate DOC-flux-dependent sensitivity indices for SR (3.62) and HR (5.12)? These numbers are not shown in Table 2. 8) Table 1, it is better to have significance test for the differences between rainy and dry season.

---

## Referee Comment (RC4) · Anonymous Referee #3 · 12 Jul 2016

General comments This study aimed to determine the relationship between the DOC transported by hydrological processes and the seasonal variation of soil respiration in a tropical rainforest. The authors measured not only the DOC flux and soil respiration, but also13C natural abundance for each DOC sources to discuss both of the production and transportation of DOC into soil. The idea is novel and worth. The writing is easy to follow. However, there are still several problems below.

(1) The authors found that there was clear seasonal variability in soil respiration, increasing in rainy season and decreasing in dry season. The variation of soil respiration strongly correlated with soil temperature, more than those with soil moisture content and water fluxes. Does this mean seasonal variation from rainy season to dry season

was clearer in soil temperature than in soil moisture content and water fluxes? Since rainy and dry season is generally defined by the amount of precipitation, it is hard to understand why the seasonal variation of soil respiration was explained by temperature, not water relating factors. The author should add the seasonal data of these explanatory factors in Fig. 2 to show how it looks like and also check the auto-correlation between them.

(2) The are no information how many locations where soil moisture content was measured. Since spatial heterogeneity of soil moisture content is very high in tropical forest ecosystem, certain amount of replicate is necessary.

(3) It is questionable whether the sensitivity of soil respiration can be compared between the different explanatory variables that has different ranges of variation. I think the range of seasonal variation have to be standardized to compare the sensitivity of SR between different variables.

Specific comments (1) Line 102, relative high: What means "relatively"? With what do you compare? (2) Line 128: It is unclear how many replicate each group has. (3) Line 155, in the soil of tropical rainforests: Reference is needed. (4) Line 158: You just mentioned the information of gas analyzer. Please explain how you measured soil respiration using the analyzer. (5) Line 298 sensitivity indices: I recommend you to explain this in the Calculation and statistics. (6) Line 422-429: This is a repeat of previous sentences.

---

## Author Comment (AC1) · 2 Aug 2016

Dear anonymous referee, We appreciate your encouragement and constructive suggestions for this research and efforts to improve this manuscript. We have carefully revised this manuscript according to your suggestions, and answered them one by one. All the details are as followed, and more details are in the revised manuscript text. Thank you and best regards! Sincerely your's: Wen-Jun Zhou Corresponding author: Yi-Ping Zhang (yipingzh@xtbg.ac.cn)

Major comments 1. there were several issues, affecting on the quality of this manuscript: The most important issue was the statistical testing: the use of one-way Anova seems to be not really appropriate for this kind of time-series data. I would

suggest using a linear mixed model with repeated measures. Missing or unbalanced data are usually no problem for this kind of analyses. Answer: Thanks for your valuable suggestion. We detected $\delta$13CDOC of every mixed samples of rainfall, throughfall, litter leachate, and soil water at 20cm depth sepeartely. We got only $\delta$13CDOC data of each kind sample for every ANALYSIS time.Tthat is mean, the data did not satisfied with the repeated measurement analyzing of the linear mixed model with repeated measures Otherwise, we just want to detect the difference between hydrological processes in $\delta$13CDOC in the rainy season and dry season separately, so one way nova analysis was used in this manuscript.

2 The second point is that, although the authors made some statistical testing, it hardly was shown anywhere. Please show the results, either in a table or incorporated into the text. Answer: Thanks for your kind suggestion. We have added statistic results in the table as below. Table2 DOC ïĄď13C dynamics along the hydrological processes (R, rainfall, TF, throughfall, LL, litter leachate) and the ïĄď13C in leaves, litter, and surface soil in the tropical rainforest at Xishuangbanna, southwest China

Season R TF LL Soil water Leaves Litter Soil (0–20 cm) (0–20 cm) Rainy season −23.9±3.3a −28.7±1.7 bc −28.1±2.7 bc −23.9±1.6 a * −32.4±0.6d −30.4±0.2cd −27.3±0.1b Dry season −23.8±1.3a −29.1±1.6 bc −28.1±1.5 bc −27.1±2.2b −32.5±0.5d −30.2±0.1cd −27.3±0.1bc R indicates rainfall, TF indicates throughfall, LL indicates litter leachate, SW20 indicates soil water at a depth of 20 cm. Different superior letters indicate significant differences between the treatments according to Lsd test (P < 0.05). *indicates the significant seasonal difference according to independent sample t test (p < 0.1)

3 The third issue was that I was missing data on soil temperature and moisture. For example, it could be easily incorporated into Figure 2, as such. Answer: We have combined the soil temperature and moisture in Figure2 as your suggestion. Thanks.

4 Finally, it was not clear to me how the authors calculated all the sensitivity indices.

Answer: Firstly, weekly soil respirations fluxes, weekly average of soil temperature (T) and soil water content (SWC), weekly water and DOC fluxes were standardized by ratio of measured value to mean value during the observation period. Secondly, linear regression equitations was used between the standardized soil respirations values and T, SWC, water and DOC fluxes respectively. Thirdly, we considered the slope of the linear regression as the sensitivity indices which showed the soil respirations variation rate with soil temperature, soil water content, water and DOC fluxes changing. More detailed comments:

1ïïjŐLines 24–28: this sentence is not easy to understand for the reader. Line 24: "role" could be changed to "effect". Line 25: "in" could be changed to "on" . Line 27: what processes do you mean? .Line 28: what do you mean by "surface soil"? Answer: Thanks, We have revised lines 24-28 according to your comments, as the following

"To better understand the effect of the dissolved organic carbon (DOC) transported by hydrological processes (throughfall, litter leachate, and surface soil water (0–20 cm)) on soil respiration in tropical rainforests, we detected the DOC flux in rainfall, throughfall, litter leachate, and surface soil water (0–20 cm), compared the seasonality of ïĄď13CDOC in each hydrological process, and ïĄď13C in leaves, litter, and surface soil, and analyzed throughfall, litter leachate, and surface soil water (0–20 cm) effect on soil respiration in a tropical rainforest in Xishuangbanna, southwest China."

2. Lines 52 and 54: first you state that laboratory studies have shown, later you write: however, most studies have been performed in the laboratory. Answer: Thanks for your kind reminding, we revised this sentence as "Laboratory studies have shown that DOC also plays a key role in SR in the surface soil (De Troyer et al., 2011, Fröberg et al., 2005, Qiao et al., 2013). However the mechanisms underlying the effects of DOC on the carbon budget and SR in the field remain unclear."

3. Line 66: do you mean both in terms of absolute and relative numbers? Answer: Yes. We have clarify it in the textto "Because of the massive rainfall in tropical rainforests,

more DOC flux is transported to the soil by throughfall and litter leachate than in other forests."

4. Line 118: do you have any additional tree data, like age or tree density? Answer: Yes, we have.and revised it as following" The dominant trees are Terminalia myriocarpa and Pometia tomentosa, which are typical tropical forest trees. Canopy height is about 45m, the land cover ratio is 100%, there are 311 species that diamater at breast height ( DBH ) is larger than 2cm (Cao et al., 1996)."

5. Line 137: how long were the tubes? Answer: The tube is about 3 meters for avoiding the disturbance from sampling on surface soil and litter layer.

6. Line 156: you removed the roots? Answer: We did not remove the roots. But we set trenched treatment before soil respiration measured 3 months, and let the died roots decomposed in the trenched treatments.

7. Line 198: how often they occurred during the dry season? Answer: Water sampling frequency depended on each hydrological progresses occurred frequency. If there was rainfall events, then we got rainfall sample in the next day, and the same to throughfall, litter leachate and soil water samples.

8. Line 221: how this was calculated? Weekly divided by 7? Answer: We calculated the weekly (7 days) water and DOC flux by summed up the daily water and DOC flux respectively.

9. Lines 221–224:from this sentence it is not entirely clear, what was compared to what Answer: To clarify the meaning, we revised this sentence to "nonlinear regression tests was used to simulate tee correlations between daily water flux and DOC concentration , between SR, HRand soil moisture, and soil temperature." 10. Line 259: is this annual average? Answer: Yes, it was. We also revised this sentence to "The highest annual interception rate was between the litter leachate and the surface soil (63.85 $\pm$ 7.98%)" . Thanks.

11. Lines 256–269: how about putting interception values into a table for better comparison? Answer: Thanks, we have filled the water and DOC flux interception values in Table 1as following. Table 1 The interception rate of the water between hydrological processes in the tropical rainforest at Xishuangbanna southwest China

. 12. Lines 272–287: somehow I could not follow all these differences from table 1 Answers: Thanks, we have added all the statistic results in this table.

13. Line 292: this is already discussion, please move it there Answers: Thanks, we have removed it.

14. Lines 304–308: it was not clear to me how you calculated the sensitivity indices Answers: Here we have recalculated it according the third referee's comments, please see the calculated details in the answer for question 4.

15. Fig.s1: what about a possible dilution effect, resulting in lower doc with more water? Answer: Yes, there were some dilution effect on DOC concentration as the follows regression equations used for the water flux and DOC concentration (Y = aebx) CTF = 48.69e−0.097x adjusted r2 = 0.3883, p = 0.002 (2) CLL = 60.93e−0.048x adjusted r2 = 0.4131, p < 0.001 (3) C sw = 6.78e−0.02048x adjusted r2 = 0.5840, p < 0.001 (4) where C TF, CLL, and Csw are the DOC concentrations (mg L−1) in the throughfall, litter leachate, and soil water (0–20 cm), respectively, and x is the water flux per day (mm).

Please also note the supplement to this comment:
http://www.biogeosciences-discuss.net/bg-2016-225/bg-2016-225-AC1-supplement.pdf
* * *
[Figure]

**Fig. 1.** Figure 2 Dynamics of soil respiration (SR) and heterotrophic respiration (HR) (a) and soil temperature at 5cm and soil water content at 10cm (b) in the tropical rainforest at Xishuangbanna, southwest

---

## Author Comment (AC2) · 2 Aug 2016

This comment is about Figure S2: why there is a gap in the temperature data? No data is presented between 18 and 21 degrees? Answer: We observed soil respiration every 2 weeks, this may lead a gap of soil temperatures during the spare time. According to the original field data, there were not data observed from 18.2 to 21.0 as the following Answer-Figure1 showed.

[Figure]

[Figure]

**Fig. 1.** Answer-Figure1 Dynamics of air and soil temperature at 5cm and 10 cm depths during soil respiration observation period in the tropical rainforest at Xishuangbanna, southwest China.

---

## Author Comment (AC3) · 2 Aug 2016

Dear anonymous referee, We deeply appreciate your considerations of our manuscript for Biogeosciences. We have carefully considered all yours comments and suggestions and made the revision accordingly. Below, we briefly introduce how we revised the manuscript. We would like to express our thankfulness to you for taking time and effort to provide such insightful comments which further improved our research paper. We look forward to receiving any further comments from you. Thank you and best regards! Yours sincerely: Wen-Jun Zhou Corresponding author: Yi-Ping Zhang (yipingzh@xtbg.ac.cn) Major comments 1 One is about the sensitivity index. We know that soil respiration increases with increasing temperature, and Q10 is widely used to de-

termine the temperature sensitivity. The authors developed similar sensitivity index for soil respiration to water fluxes, DOC fluxes and soil water content. I believe that these kinds of sensitivity index are useful when comparing them among different sites, as is the Q10. However, I don't think we can compare among the temperature sensitivity, soil water content sensitivity, water flux sensitivity, DOC flux sensitivity within the same site, because they are different parameters and the units for each parameter are different. Thus the authors need to provide the rationales for these comparisons, otherwise the conclusions stated by lines 35 to 38 are different to stand. Answer: Thanks for your significant comments and suggestion on the sensitivity indices. In order to be able to evaluate the sensitivity of soil respiration towards soil temperature, soil water content, water and DOC fluxes to soil respirations in tropical , we have standardized all the parameters by the ratio of measured value to the means of the observation period. And consider the slop of linear regression between soil respiration and soil temperature, soil water content and water and DOC fluxes as the sensitivity indies. In this way, we compared sensitivity of soil respirations to all of these parameters which originally have different unit.

2 The other concern is about the importance of DOC. DON input from throughfall accounted for about 7% of the net ecosystem C exchange. However, it may be even minor when compared to soil respiration. So it needs to not overstate the importance of DOC in C budget. The phrase of "key" in the abstract (line 32) and throughout the manuscript may be not proper, to my point of view. It may be better to use "important" instead "key".

Answer: thanks for your advise, we use the "important" for all the description of DOC role the text.

Specific comments:

1) Line 96, in a tropical forest;

Answer: This sentence was revised to "Our study was performed in a tropical rainforest

at Xishuangbanna in southwest China, on the northern edge of a tropical region." As your suggestion.

2) Line 124, how large is your study plot?

Answer: We have added the plot area as the following description "At the study plot (a 23.4 ha catchment)".

3) Line 127, "the" may be not necessary;

Answer: Thanks for your careful suggestion, we have deleted "the" in this sentence and revised to "To sample throughfall, litter leachate, and soil water (20 cm depth), four groups of replicate collectors were set for each of these measurements."

4) Line 179, 2 to 6 mg? what standards were used to calibrate the measured values for plant and soil samples, as well as for DOC samples? Answer: We revised the sample weights to"1.00-3.00 mg plant samples and 10-40 mg soil sample dried and sieved through 100 mesh size "according to the analyzing original records. We used low organic soil standard (CatNo.B2153) for soil and DOC and wheat flour standard (CatNo.B2157) for plant sample determination of $\delta$13C respectively. The standards were certified in Organic Analytical Standard (IAS/OAS) at Elemental Microanalysis Ltd(Oakhampton, Devon, UK).

5) Line 291, the contribution of HR to total soil respiration was 72%, which is in higher than many reports for forests? Is this normal? Answer: Hanson et al (2000) showed heterotrophic respiration is about 54% of total soil respiration globally. The ratio is 30-83% of the total soil respiration in temperature and tropical forests(Behera et al,1990; Epron et al 1999, Tomotsune et al, 2013) and 7-50% in boreal forest (Matsushita,2015). So HR contributed 72% of the total soil respiration of this research is in the higher level compared to the research in the global. Behara, N., Joshi ,S. K., Pati, D. P.: Root contribution to total soil metabolism in a tropical forest soil from Orissa, India, Forest Ecology and Management. 36: 125–134, 1990. Epron, D., Farque, L., Lucot, E.,

Badot, P-M.: Soil CO2 efflux in a beech forest: the contribution of root respiration, Ann For Sci. 56: 289-295,1999. Hanson, P., Edwards, N., Garten, C. & Andrews, J. Separating root and soil microbial contributions to soil respiration: a review of methods and observations, Biogeochemistry. 48, 115–146,2000. Matsushita,K., Tomotsune, M., Sakamaki, Y., Koizumi,H.: Effects of management treatments on the carbon cycle of a cool-temperate broad-leaved deciduous forest and its potential as a bioenergy source, Ecological Research. 30(2): 293-302,2015. Tomotsune, M., Yoshitake, S., Watanabe, S., Koizumi, H. (2013). Separation of root and heterotrophic respiration within soil respiration by trenching, root biomass regression, and root excising methods in a cool-temperate deciduous forest in Japan. Ecological Research, 28, 259-269.

6) Line 299, sensitivity of soil respiration to soil moisture has not shown in Fig S2. Answer: Thanks for your reminder, we added this in FigS2.

7) Line 309-310, how did you calculate DOC-flux-dependent sensitivity indices for SR (3.62) and HR (5.12)? These numbers are not shown in Table 2. Answer: We calculated all the hydrological processes DOC-flux-dependent sensitivity as the average±standadr deviation for both SR and HR.

8) Table 1, it is better to have significance test for the differences between rainy and dry season. Answer: Thanks, We have added the statistic results in the table which has been changed to Table 2 between rainy and dry season and between hydrological processes.

Table 2 DOC ïĄď13C dynamics along the hydrological processes (R, rainfall, TF, throughfall, LL, litter leachate) and the ïĄď13C in leaves, litter, and surface soil in the tropical rainforest at Xishuangbanna, southwest China

Season R TF LL Soil water (0–20 cm) Leaves Litter Soil (0–20 cm) Rainy season −23.9±3.3a −28.7±1.7 bc −28.1±2.7 bc −23.9±1.6 a * −32.4±0.6d −30.4±0.2cd −27.3±0.1b Dry season −23.8±1.3a −29.1±1.6 bc −28.1±1.5 bc −27.1±2.2b −32.5±0.5d −30.2±0.1cd −27.3±0.1bc

R indicates rainfall, TF indicates throughfall, LL indicates litter leachate, SW20 indicates soil water at a depth of 20 cm. Different superior letters indicate significant differences between the treatments according to Lsd test (P < 0.05). *indicates the significant seasonal difference according to independent sample t test (p < 0.1)

Please also note the supplement to this comment:
http://www.biogeosciences-discuss.net/bg-2016-225/bg-2016-225-AC3-supplement.pdf

[Figure]

**Fig. 1.** Figure S2 Correlation between soil temperature and soil water content of CO2 from eddy flux tower explained during soil respiration observation plot from Feb. 2008 to Jan. 2009 (a), soil respiration a

---

## Author Response (AR1)

Dear editor and reviewers:

We appreciate your diligent review of our manuscript and the comments to the purposes. We believe your question is an appropriate and critical comment to improve this manuscript.

We have carefully read your comments and learned a lot from them.

We believe that editor and reviewers' comments on methods, statistics, tables and figures and language through the text are very important to improve this manuscript. We have had cautious answers and improved all the parts according your advices that help us further strengthen the paper. Please find details below and in the revised manuscript as well.

After these, we believe we have a pleasant communication with editor and reviewers about the manuscript, hope your further comments on this study.

Let us say thank you for your hard work again!

Sincerely yours,

Wen–Jun Zhou, Yi–Ping Zhang, Li-Qing Sha and all the co-authors

**#1 Answer to the first referee**

**Major comments**

1. there were several issues, affecting on the quality of this manuscript: The most important issue was the statistical testing: the use of one-way Anova seems to be not really appropriate for this kind of time-series data. I would suggest using a linear mixed model with repeated measures. Missing or unbalanced data are usually no problem for this kind of analyses.

*Answer: Thanks for your valuable suggestion.*

*We detected $\delta^{13}C_{DOC}$ of every mixed samples of rainfall, throughfall, litter leachate, and soil water at 20cm depth sepeartely. We got only $\delta^{13}C_{DOC}$ data of each kind sample for every ANALYSIS time.Tthat is mean, the data did not satisfied with the repeated measurement analyzing of the linear mixed model with repeated measures Otherwise, we just want to detect the difference between hydrological processes in $\delta^{13}C_{DOC}$ in the rainy season and dry season separately, so one way nova analysis was used in this manuscript.*

2. The second point is that, although the authors made some statistical testing, it hardly was shown anywhere. Please show the results, either in a table or incorporated into the text.

*Answer: Thanks for your kind suggestion. We have added statistic results in the table as below.*

Table 2 DOC $\delta^{13}C$ dynamics along the hydrological processes (R, rainfall, TF, throughfall, LL, litter leachate) and the $\delta^{13}C$ in leaves, litter, and surface soil in the tropical rainforest at Xishuangbanna, southwest China

| Season | R | TF | LL | Soil water (0–20 cm) | Leaves | Litter | Soil (0–20 cm) |
|---|---|---|---|---|---|---|---|
| Rainy season | −23.9±3.3[a] | −28.7±1.7[bc] | −28.1±2.7[bc] | −23.9±1.6[a] * | −32.4±0.6[d] | −30.4±0.2[cd] | −27.3±0.1[b] |
| Dry season | −23.8±1.3[a] | −29.1±1.6[bc] | −28.1±1.5[bc] | −27.1±2.2[b] | −32.5±0.5[d] | −30.2±0.1[cd] | −27.3±0.1[bc] |

R indicates rainfall, TF indicates throughfall, LL indicates litter leachate, SW20 indicates soil water at a depth of 20 cm.

Different superior letters indicate significant differences between the treatments according to Lsd test (P < 0.05).

*indicates the significant seasonal difference according to independent sample t test (p < 0.1)

The third issue was that I was missing data on soil temperature and moisture. For example, it could be easily incorporated into Figure 2, as such.

*Answer: We have combined the soil temperature and moisture in Figure2 as your suggestion. Thanks.*

[Figure]

*Figure 2 Dynamics of soil respiration (SR) and heterotrophic respiration (HR) (a) and soil temperature at 5cm and soil water content at 10cm (b) in the tropical rainforest at Xishuangbanna, southwest China.*

*The shaded area indicates the rainy season.*

Finally, it was not clear to me how the authors calculated all the sensitivity indices.

*Answer: Firstly, weekly soil respirations fluxes, weekly average of soil temperature (T) and soil water content (SWC), weekly water and DOC fluxes were standardized by ratio of measured value to mean value during the observation period. Secondly, linear regression equitations was used*

*between the standardized soil respirations values and T, SWC, water and DOC fluxes respectively. Thirdly, we considered the slope of the linear regression as the sensitivity indices which showed the soil respirations variation rate with soil temperature, soil water content, water and DOC fluxes changing.*

**More detailed comments:**

1) Lines 24–28: this sentence is not easy to understand for the reader. Line 24: "role" could be changed to "effect". Line 25: "in" could be changed to "on" . Line 27: what processes do you mean? .Line 28: what do you mean by "surface soil"?

*Answer: Thanks, We have revised lines 24-28 according to your comments, as the following "To better understand the effect of the dissolved organic carbon (DOC) transported by hydrological processes (throughfall, litter leachate, and surface soil water (0–20 cm)) on soil respiration in tropical rainforests, we detected the DOC flux in rainfall, throughfall, litter leachate, and surface soil water (0–20 cm), compared the seasonality of $\delta^{13}C_{DOC}$ in each hydrological process, and $\delta^{13}C$ in leaves, litter, and surface soil, and analyzed throughfall, litter leachate, and surface soil water (0–20 cm) effect on soil respiration in a tropical rainforest in Xishuangbanna, southwest China."*

2) Figure S2: why there is a gap in the temperature data? No data
is presented between 18 and 21 degrees?

*Answer: We observed soil respiration every 2 weeks, this may lead a gap of soil temperatures during the spare time. According to the original field data, there were not data observed from 18.2 to 21.0 as the following Answer-Figure1 showed.*

[Figure]

*Answer-Figure1 Dynamics of air and soil temperature at 5cm and 10 cm depths during soil respiration observation period in the tropical rainforest at Xishuangbanna, southwest China.*

**#2 Answer to the second refree**

**Major comments**

One is about the sensitivity index. We know that soil respiration increases with increasing temperature, and Q10 is widely used to determine the temperature sensitivity. The authors developed similar sensitivity index for soil respiration to water fluxes, DOC fluxes and soil water content. I believe that these kinds of sensitivity index are useful when comparing them among different sites, as is the Q10. However, I don't think we can compare among the temperature sensitivity, soil water content sensitivity, water flux sensitivity, DOC flux sensitivity within the same site, because they are different parameters and the units for each parameter are different. Thus the authors need to provide the rationales for these comparisons, otherwise the conclusions stated by lines 35 to 38 are different to stand.

*Answer: Thanks for your significant comments and suggestion on the sensitivity indices. In order to be able to evaluate the sensitivity of soil respiration towards soil temperature, soil water content, water and DOC fluxes to soil respirations in tropical , we have standardized all the parameters by the ratio of measured value to the means of the observation period. And consider the slop of linear regression between soil respiration and soil temperature, soil water content and water and DOC fluxes as the sensitivity indies. In this way, we compared sensitivity of soil respirations to all of these parameters which originally have different unit.*

The other concern is about the importance of DOC. DON input from throughfall accounted for about 7% of the net ecosystem C exchange. However, it may be even minor when compared to soil respiration. So it needs to not overstate the importance of DOC in C budget. The phrase of "key" in the abstract (line 32) and throughout the manuscript may be not proper, to my point of view. It may be better to use "important" instead "key".

*Answer: thanks for your advise, we use the "important" for all the description of DOC role the text.*

**Specific comments:**

1) Line 96, in a tropical forest;

*Answer: This sentence was revised to "Our study was performed in a tropical rainforest at Xishuangbanna in southwest China, on the northern edge of a tropical region." As your suggestion.*

2) Line 124, how large is your study plot?

*Answer: We have added the plot area as the following description "At the study plot (a 23.4 ha catchment)".*

3) Line 127, "the" may be not necessary;

*Answer: Thanks for your careful suggestion, we have deleted "the" in this sentence and revised to "To sample throughfall, litter leachate, and soil water (20 cm depth), four groups of replicate collectors were set for each of these measurements."*

4) Line 179, 2 to 6 mg? what standards were used to calibrate the measured values for plant and soil samples, as well as for DOC samples?

*Answer: We revised the sample weights to"1.00-3.00 mg plant samples and 10-40 mg soil sample dried and sieved through 100 mesh size "according to the analyzing original records.*

*We used low organic soil standard (CatNo.B2153) for soil and DOC and wheat flour standard (CatNo.B2157) for plant sample determination of $\delta^{13}C$ respectively. The standards were certified in Organic Analytical Standard (IAS/OAS) at Elemental Microanalysis Ltd(Oakhampton, Devon, UK).*

5) Line 291, the contribution of HR to total soil respiration was 72%, which is in higher than many reports for forests? Is this normal?

*Answer: Hanson et al (2000) showed heterotrophic respiration is about 54% of total soil respiration*

*globally. The ratio is 30-83% of the total soil respiration in temperature and tropical forests(Behera et al,1990; Epron et al 1999, Tomotsune et al, 2013) and 7-50% in boreal forest (Matsushita,2015). So HR contributed 72% of the total soil respiration of this research is in the higher level compared to the research in the global.*

*Behara, N., Joshi ,S. K., Pati, D. P.: Root contribution to total soil metabolism in a tropical forest soil from Orissa, India, Forest Ecology and Management. 36: 125–134, 1990.*

*Epron, D., Farque, L., Lucot, E., Badot, P-M.: Soil $CO_2$ efflux in a beech forest: the contribution of root respiration, Ann For Sci. 56: 289-295,1999.*

*Hanson, P., Edwards, N., Garten, C. & Andrews, J. Separating root and soil microbial contributions to soil respiration: a review of methods and observations, Biogeochemistry. 48, 115–146,2000.*

*Matsushita,K., Tomotsune, M., Sakamaki, Y., Koizumi,H.: Effects of management treatments on the carbon cycle of a cool-temperate broad-leaved deciduous forest and its potential as a bioenergy source, Ecological Research. 30(2): 293-302,2015.*

*Tomotsune, M., Yoshitake, S., Watanabe, S., Koizumi, H. (2013). Separation of root and heterotrophic respiration within soil respiration by trenching, root biomass regression, and root excising methods in a cool-temperate deciduous forest in Japan. Ecological Research, 28, 259-269.*

6) Line 299, sensitivity of soil respiration to soil moisture has not shown in Fig S2.

*Answer: Thanks for your reminder, we added this in FigS2 as following*

[Figure]

Figure S2 Correlation between soil temperature and soil water content of $CO_2$ from eddy flux tower explained during soil respiration observation plot from Feb. 2008 to Jan. 2009 (a), soil respiration and temperature at 5 cm depth (b), and soil water content at 10 cm depth (c) in the tropical rainforest at Xishuangbanna, southwest China

7) Line 309-310, how did you calculate DOC-flux-dependent sensitivity indices for SR (3.62) and HR (5.12)? These numbers are not shown in Table 2.

*Answer: We calculated all the hydrological processes DOC-flux-dependent sensitivity as the mean ± standadr deviation for both SR and HR accdording to the new methods.*

8) Table 1, it is better to have significance test for the differences between rainy and dry season.

*Answer: Thanks, We have added the statistic results in the table which has been changed to Table 2 between rainy and dry season and between hydrological processes.*

Table 2 DOC $\delta^{13}$C dynamics along the hydrological processes (R, rainfall, TF, throughfall, LL, litter leachate) and the $\delta^{13}$C in leaves, litter, and surface soil in the tropical rainforest at Xishuangbanna, southwest China

| Season | R | TF | LL | Soil water(0–20 cm) | Leaves | Litter | Soil (0–20 cm) |
|---|---|---|---|---|---|---|---|
| Rainy season | –23.9±3.3[a] | –28.7±1.7[bc] | –28.1±2.7[bc] | –23.9±1.6[a] * | –32.4±0.6[d] | –30.4±0.2[cd] | –27.3±0.1[b] |
| Dry season | –23.8±1.3[a] | –29.1±1.6[bc] | –28.1±1.5[bc] | –27.1±2.2[b] | –32.5±0.5[d] | –30.2±0.1[cd] | –27.3±0.1[bc] |

R indicates rainfall, TF indicates throughfall, LL indicates litter leachate, SW20 indicates soil water at a depth of 20 cm.

Different superior letters indicate significant differences between the treatments according to Lsd test ($P < 0.05$).

*indicates the significant seasonal difference according to independent sample t test ($p < 0.1$)

2. Lines 52 and 54: first you state that laboratory studies have shown, later you write: however, most studies have been performed in the laboratory.

*Answer: Thanks for your kind reminding, we revised this sentence as "Laboratory studies have shown that DOC also plays a key role in SR in the surface soil (De Troyer et al., 2011, Fröberg et*

*al., 2005, Qiao et al., 2013). However the mechanisms underlying the effects of DOC on the*

*carbon budget and SR in the field remain unclear."*

3. Line 66: do you mean both in terms of absolute and relative numbers?

*Answer: Yes. We have clarify it in the textto "Because of the massive rainfall in tropical*

*rainforests, more DOC flux is transported to the soil by throughfall and litter leachate than in*

*other forests."*

4. Line 118: do you have any additional tree data, like age or tree density?

*Answer: Yes, we hav*e.and revised it as following" The dominant trees are Terminalia myriocarpa and

Pometia tomentosa, which are typical tropical forest trees. Canopy height is about 45m, the land cover ratio is 100%, there are 311 species that diamater at breast height ( DBH ) is larger than 2cm (Cao et al.,

1996)."

5. Line 137: how long were the tubes?

*Answer: The tube is about 3 meters for avoiding the disturbance from sampling on surface soil*

*and litter layer.*

6. Line 156: you removed the roots?

*Answer: We did not remove the roots. But we set trenched treatment before soil respiration*

*measured 3 months, and let the died roots decomposed in the trenched treatments.*

7. Line 198: how often they occurred during the dry season?

*Answer: Water sampling frequency depended on each hydrological progresses occurred frequency.*

*If there was rainfall events, then we got rainfall sample in the next day, and the same to*

*throughfall, litter leachate and soil water samples.*

8. Line 221: how this was calculated? Weekly divided by 7?

*Answer: We calculated the weekly (7 days) water and DOC flux by summed up the daily water*

*and DOC flux respectively.*

9. Lines 221–224:from this sentence it is not entirely clear, what was compared to what

*Answer*: *To clarify the meaning, we revised this sentence to "nonlinear regression tests was used to simulate tee correlations between daily water flux and DOC concentration , between SR, HRand soil moisture, and soil temperature."*

10. Line 259: is this annual average?

*Answer: Yes, it was.*

*We also revised this sentence to "The highest annual interception rate was between the litter leachate and the surface soil (63.85 ± 7.98%)" . Thanks.*

11. Lines 256–269: how about putting interception values into a table for better comparison?

*Answer*: *Thanks, we have filled the water and DOC flux interception values in Table 1as following.*

**Table 1 The interception rate of the water between hydrological processes in the tropical rainforest at Xishuangbanna southwest China**

|  | Interceptation | Annual | Rainy season | Dry season |
|---|---|---|---|---|
| Water flux | Between TF and R | 53.9±11.7 | 43.1±2.7 | 41.3±14.8 |
|  | Between LL and TF | 33.9±6.6 | 33.9±9.8 | 34.1±27.6 |
|  | Between SW20cm and LL | 63.8±8.0 | 62.2±15.1 | 81.6±23.3 |
| DOC flux | Between TF and R | 137.0±19.9 | 182.0±16.0 | 170.8±7.8 |
|  | Between LL and TF | 1.1±17.0 | 16.1±9.4 | 12.7±4.3 |
|  | Between SW20cm and LL | -96.7±4.4 | -93.9±2.6 | -94.4±1.2 |

.

12. Lines 272–287: somehow I could not follow all these differences from table 1

*Answers*: *Thanks, we have added all the statistic results in this table.*

13. Line 292: this is already discussion, please move it there

*Answers: Thanks, we have removed it.*

14. Lines 304–308: it was not clear to me how you calculated the sensitivity indices

*Answers: Here we have recalculated it according the third referee's comments, please see the calculated details in the answer for question 4.*

15. Fig.s1: what about a possible dilution effect, resulting in lower doc with more water?

*Answer: Yes, there were some dilution effect on DOC concentration as the follows regression equations used for the water flux and DOC concentration ($Y = ae^{bx}$)*

$$C_{TF} = 48.69e^{-0.097x} \quad adjusted\ r^2 = 0.3883,\ p = 0.002 \qquad (2)$$

$$C_{LL} = 60.93e^{-0.048x} \quad adjusted\ r^2 = 0.4131,\ p < 0.001 \qquad (3)$$

$$C_{sw} = 6.78e^{-0.02048x} \qquad adjusted\ r^2 = 0.5840,\ p < 0.001 \qquad (4)$$

*where $C_{TF}$, $C_{LL}$, and $C_{sw}$ are the DOC concentrations (mg $L^{-1}$) in the throughfall, litter leachate, and soil water (0–20 cm), respectively, and x is the water flux per day (mm).*

**#3 Answer for the third refree**

**Major comments**

The authors found that there was clear seasonal variability in soil respiration, increasing in rainy season and decreasing in dry season. The variation of soil respiration strongly correlated with soil temperature, more than those with soil moisture content and water fluxes. Does this mean seasonal variation from rainy season to dry season was clearer in soil temperature than in soil moisture content and water fluxes? Since rainy and dry season is generally defined by the amount of precipitation, it is hard to understand why the seasonal variation of soil respiration was explained by temperature, not water relating factors. The author should add the seasonal data of these explanatory factors in Fig. 2 to show how it looks like and also check the auto-correlation between them.

*Answer: 1)Thanks, we have added the dynamics figure of soil temperature at 5cm and soil water content at 10cm depth as Fig2b.*

[Figure]

*Figure 2* *Dynamics of soil respiration (SR) and heterotrophic respiration (HR) (a) and soil temperature at 5cm and soil water content at 10cm (b) in the tropical rainforest at Xishuangbanna, southwest China. The shaded area indicates the rainy season.*

*2) We have checked the correlation between soil temperature at 5cm depth (T5) and soil water content at 10 cm (SWC10) showed SWC10 = 1.38+1.00 T5 ($r^2$ = 0.3293, p < 0.0001), this indicated soil temperature at 5cm depth explained 32.93% soil water content. This showed soil temperature was not all in covariance with soil water content(Fig 2b). This can induce that soil respiration is in the similar dynamic with soil temperature. While with the water input, soil microbe will be influenced by soil water content and DOC- the more activity C fraction, thus, water input also contributed to the good correlation between soil temperature and soil respiration which can proved by table 3.*

The are no information how many locations where soil moisture content was measured. Since spatial heterogeneity of soil moisture content is very high in tropical forest ecosystem, certain amount of replicate is necessary.

*Answer: Thanks, we have detected 30 chambers(5 trench plots×3 chambers + 5 control plots ×3 chambers) soil moisture and soil temperature.*

It is questionable whether the sensitivity of soil respiration can be compared between the different explanatory variables that has different ranges of variation. I think the range of seasonal variation have to be standardized to compare the sensitivity of

SR between different variables.

*Answer: First of all, thanks for your question of great insight. We have standardized all the parameters and recalculated the sensitive indices as the following steps:    Firstly, weekly soil respirations fluxes, weekly average of soil temperature (T) and soil water content (SWC), weekly water and DOC fluxes were standardized by the ratio of measured valueto the mean value during the observation period. Secondly, linear regression equitation was used between the standardized soil respirations values and T, SWC, water and DOC fluxes respectively. Thirdly, we considered the slope of the linear regression as the sensitivity indices which showed the soil respirations variation rate with soil temperature, soil water content, water and DOC fluxes changing.*

*We also have explained this in the caculation and statistic in method.*

**Specific comments**

1) Line 102, relative high: What means "relatively"? With do you compare?

*Answer: Thanks, this sentence is confused, so we revised it to "We hypothesized that throughfall and litter leachate DOC flux are important in carbon budget" with more clear expression.*

2) Line 128: It is unclear how many replicate each group has.

*Answer: Each group has a throughfall, a litter leachate, and a soil water (20 cm depth) collectors in each group, so there are 4 replicates for every hydrological processes. We revised this sentence to "To sample throughfall, litter leachate, and soil water (20 cm depth), four groups of replicate collectors were set for each of these measurements" to "There were four replicates of throughfall, litter leachate, and soil water (20 cm depth) respectively. All the collectors were set around the eddy flux tower randomly."*

(3) Line 155, in the soil of tropical rainforests: Reference is needed.

*Answer: Thanks, we have added the reference in the text and the reference list.*

(4) Line 158: You just mentioned the information of gas analyzer. Please explain how you measured soil respiration using the analyzer.

*Answer: The soil respiration was measured using a Li-820 system (Li-Cor Inc., Lincoln, NE, USA), which consisted of an infrared gas analyzer with a polyvinyl chloride chamber(diameter of 15cm and height of 15.0 cm). A polyvinyl chloride collar(diameter of 15cm an height of 5cm) was installed in the forest floor to a depth of ~3 cm. All the leaf litter and small branches were left in the collar. Soil respirations were detected from 09:00 to 14:00 local time when was taken to represent respiration in that day (Sha et al. 2005, Yao et al. 2011).*

*We have revised it in the method.*

(5) Line 298 sensitivity indices: I recommend you to explain this in the Calculation and statistics.

*Answer:* Thanks, we have added it in the calculation and statistics details.

 "*In order to evaluate the variation of soil temperature, soil water content, water and DOC fluxes to soil respirations in tropical, we have standardized all the parameters by the measured value sub the means of the observation period. And consider the slope of linear regression between soil respiration and soil temperature, soil water content and water and DOC fluxes as the sensitivity indices. In this way, we compared sensitivity of soil respirations to all of these parameters.*"

(6) Line 422-429: This is a repeat of previous sentences.

*Answer: We have deleted these lines.*

[revised manuscript text omitted]
 | $-23.9 \pm 3.3^{a}$ | $-28.7 \pm 1.7^{bc}$ | $-28.1 \pm 2.7^{bc}$ | $-23.9 \pm 1.6^{a}$ * | $-32.4 \pm 0.6^{d}$ | $-30.4 \pm 0.2^{cd}$ | $-27.3 \pm 0.1^{b}$ |
| Dry season | $-23.8 \pm 1.3^{a}$ | $-29.1 \pm 1.6^{bc}$ | $-28.1 \pm 1.5^{bc}$ | $-27.1 \pm 2.2^{b}$ | $-32.5 \pm 0.5^{d}$ | $-30.2 \pm 0.1^{cd}$ | $-27.3 \pm 0.1^{bc}$ |

R indicates rainfall, TF indicates throughfall, LL indicates litter leachate, SW20 indicates soil water at a depth of 20 cm.

Different superior letters indicate significant differences between the treatments according to Lsd test (P < 0.05).

*indicates the significant seasonal difference according to independent sample t test (p < 0.1)

Table 23

|  |  | SR | | | | HR | | | |
|---|---|---|---|---|---|---|---|---|---|
|  |  | a | b | $R^2$ | p | a | b | $R^2$ | p |
| DOC flux | T | 0.56 | 0.54 | 0.987 | <0.001 | 0.46 | 0.64 | 0.982 | <0.001 |
|  | SWC | 0.65 | 0.41 | 0.558 | <0.001 | 0.53 | 0.52 | 0.568 | <0.001 |
|  | R | 2.31 | -1.17 | 0.423 | <0.001 | 1.86 | -0.74 | 0.425 | <0.001 |
|  | TF | 2.36 | -1.25 | 0.429 | <0.001 | 1.91 | -0.83 | 0.413 | <0.001 |
|  | LL | 2.71 | -1.57 | 0.355 | <0.001 | 2.21 | -1.10 | 0.366 | <0.001 |
|  | SW20 | 3.57 | -2.23 | 0.227 | <0.001 | 2.91 | -1.62 | 0.240 | <0.001 |
|  | LL-SW20 | 2.66 | -1.53 | 0.352 | <0.001 | 2.17 | -1.07 | 0.363 | <0.001 |
| Water flux | R | 2.42 | -1.35 | 0.323 | <0.001 | 1.96 | -0.92 | 0.331 | <0.001 |
|  | TF | 2.55 | -1.44 | 0.316 | <0.001 | 2.06 | -0.99 | 0.323 | <0.001 |
|  | LL | 3.02 | -1.83 | 0.301 | <0.001 | 2.46 | -1.31 | 0.312 | <0.001 |
|  | SW20 | 3.70 | -2.34 | 0.166 | <0.001 | 3.02 | -1.71 | 0.178 | <0.001 |
|  | LL-SW20 | 2.64 | -1.54 | 0.257 | <0.001 | 2.14 | -1.08 | 0.267 | <0.001 |

|  | Parameters |  | a | b | $r^2$ | P | Sensitivity index |
|---|---|---|---|---|---|---|---|
| Water flux | TF | SR | 354.80 | 0.0064 | 0.4271 | <0.0001 | 1.07 |
|  |  | HR | 252.17 | 0.0075 | 0.4178 | <0.0001 | 1.08 |
|  | LL | SR | 380.19 | 0.0058 | 0.2469 | <0.0001 | 1.06 |
|  |  | HR | 273.98 | 0.0068 | 0.2556 | <0.0001 | 1.07 |
|  | LL-SW20 | SR | 375.65 | 0.0095 | 0.2525 | <0.0001 | 1.10 |
|  |  | HR | 269.67 | 0.0112 | 0.2469 | <0.0001 | 1.12 |
|  | SW20 | SR | 404.49 | 0.0062 | 0.1194 | <0.0001 | 1.06 |
|  |  | HR | 294.92 | 0.0073 | 0.119 | <0.0001 | 1.08 |
| DOC flux | TF | SR | 341.71 | 0.0441 | 0.4681 | <0.0001 | 1.55 |
|  |  | HR | 240.71 | 0.0524 | 0.459 | <0.0001 | 1.69 |
|  | LL | SR | 355.26 | 0.0316 | 0.4061 | <0.0001 | 1.37 |
|  |  | HR | 251.47 | 0.0405 | 0.3971 | <0.0001 | 1.50 |
|  | LL-SW20cm | SR | 355.00 | 0.0336 | 0.4021 | <0.0001 | 1.40 |
|  |  | HR | 251.47 | 0.0402 | 0.3971 | <0.0001 | 1.49 |
|  | SW20 | SR | 392.50 | 0.2318 | 0.2053 | <0.0001 | 10.16 |
|  |  | HR | 284.09 | 0.276 | 0.276 | <0.0001 | 15.80 |
|  | T5 | SR | 46.37 | 0.11 | 0.89 | <0.0001 | 3.00 |
|  |  | HR | 18.90 | 0.14 | 0.84 | <0.0001 | 4.06 |
|  | swc | SR | 153.30 | 0.0274 | 0.3756 | 0.0004 | 1.32 |
|  |  | HR | 96.85 | 0.0314 | 0.3199 | 0.0013 | 1.37 |

Equations used to calculate the sensitivity indices: sensitivity index = $e^{(10b)}a$, where b is the parameter constant of the regression equation for standrised SRsoil respirations, and for soil temperature, soil water content, water flux, and DOC flux: $Y = ax+be^{bx}$, where Y is the standrised soil respiration rate, and x is standrised soil temperature, soil water content, water flux, or DOC flux.

T indicates soil temperature at a depth of 5 cm, SWC indicates soil water content,TF indicates throughfall, LL indicates litter leachate, SW20 indicates soil water at a depth of 20 cm, LL- SW20cm indicates the difference between litter leachate and soil water at a depth of 20 cm,. T5 indicates soil temperature at a depth of 5 cm, SWC indicates soil water content.

Figure 1

[Figure]

Figure 2

[Figure]

---

## Author Response (AR2)

1 Dear editor,

2 We are grateful for your consideration and the concise suggestions on our manuscript.

3 Here we have revised the manuscript according to your advises thoroughly. And all the details are 4

as following.

5 6

7 1. Please do not use "R" to denote rainfall. This is confusing with regression coefficient. Also, 8 please use the same symbol in text and table. Namely, in the present manuscript, you used "r" in text and "R" in tables for regression coefficients. Instead, for example, I suggest using "P" for 9 10 precipitation and "R" for regression coefficient through the manuscript.

11

12 Answer: Thanks, we have revised the denote of rainfall R to P in tables and figures.

13

2. In general, fluxes should have units of mass per unit area per unit time (e.g., g m-2 s-1 or mol 14 15 m-2 h-1). However, in your manuscript, many flux terms lack necessary dimensions. For example, in Figure 1, DOC flux may have units of (kg C ha-1 yr-1). Please check all figures and tables, 16 17 including those in Supplementary materials. 18 Answer: We have added the unit of Table 1 and 2. But for the Figure 1, we just show the annual 19 and rainy season flux of water and DOC flux, so the unit time were not shown and indicted the 20 21 DOC flux in the text as (kg C ha-1 yr-1). Thanks 22 23 24 3. The surface CO2 flux in Figure 2a seems too high, if the unit of y-axis (mg CO2 m-2 s-1) is 25 correct. Please check.

26

27 Answer: Thanks for your kind check. We have checked the data and revised the unit to (mg CO2  $m^{-2} h^{-1}$ ) in Figure 2 and Figure S2. 28

29

30 Please find our revised details in the man scrip with check tracks and the clear ms.. Hope our 1

- 31 revised version is suitable for publication. Please contact us without hesitation when you have any
- 32 question about this manuscript.
- 33 Best regards!
- 34 Sincerely yours,
- 35 Wen–Jun Zhou, Yi–Ping Zhang, Li-Qing Sha and all the co-authors
- 36

- Hydrologically transported dissolved organic carbon influences soil respiration in a tropical
   rainforest
- 39 Running title: DOC influences soil respiration
- 40 Author: W.-J. Zhou1,2,3, H.-Z. Lu1,2,3, Y.-P. Zhang1,2\*, L.-Q. Sha1,2\*, D. Schaefer1,2, Q.-H. Song1,2,3,
- 41 Y. Deng1,2, X.-B. Deng1,2

[revised manuscript text omitted]
   |       | RP           | TF                      | LL                      | Soil water          | Leaves              | Litter             | Soil                    | 带格式表格 |  |
|          |       |                     |                         |                         | (0-20  cm)          |                     |                    | (0-20  cm)              |       |  |
|          |       |                     |                         |                         | % 0          |                     |                    |                         |       |  |
| Rainy se | eason | $-23.9\pm3.3^{a}$   | $-28.7\pm1.7^{bc}$      | -28.1±2.7 bc | $-23.9\pm1.6^{a}$ * | $-32.4\pm0.6^{d}$   | $-30.4\pm0.2^{cd}$ | −27.3±0.1 b  |       |  |
| Dry seas | son   | $-23.8 \pm 1.3^{a}$ | $-29.1 \pm \! 1.6^{bc}$ | $-28.1 \pm \! 1.5^{bc}$ | $-27.1\pm2.2^{b}$   | $-32.5 \pm 0.5^{d}$ | $-30.2\pm0.1^{cd}$ | −27.3±0.1 bc |       |  |

621 R-P indicates rainfall, TF indicates throughfall, LL indicates litter leachate, SW20 indicates soil water

622 at a depth of 20 cm.

623 Different superior letters indicate significant differences between the treatments according to Lsd test

624 (P < 0.05).

 $\label{eq:constraint} \textbf{625} \qquad \text{``indicates the significant seasonal difference according to independent sample t test (p < 0.1)}$

**626 Table 3**

|            |           | SR   |       |       |         | HR   |       |       |         |
|------------|-----------|------|-------|-------|---------|------|-------|-------|---------|
|            |           | а    | b     | $R^2$ | р       | а    | b     | $R^2$ | р       |
|            | Т         | 0.56 | 0.54  | 0.987 | < 0.001 | 0.46 | 0.64  | 0.982 | < 0.001 |
|            | SWC       | 0.65 | 0.41  | 0.558 | < 0.001 | 0.53 | 0.52  | 0.568 | < 0.001 |
| DOC flux   | RP | 2.31 | -1.17 | 0.423 | < 0.001 | 1.86 | -0.74 | 0.425 | < 0.001 |
|            | TF        | 2.36 | -1.25 | 0.429 | < 0.001 | 1.91 | -0.83 | 0.413 | < 0.001 |
|            | LL        | 2.71 | -1.57 | 0.355 | < 0.001 | 2.21 | -1.10 | 0.366 | < 0.001 |
|            | SW20      | 3.57 | -2.23 | 0.227 | < 0.001 | 2.91 | -1.62 | 0.240 | < 0.001 |
|            | LL-SW20   | 2.66 | -1.53 | 0.352 | < 0.001 | 2.17 | -1.07 | 0.363 | < 0.001 |
| Water flux | RP | 2.42 | -1.35 | 0.323 | < 0.001 | 1.96 | -0.92 | 0.331 | < 0.001 |
|            | TF        | 2.55 | -1.44 | 0.316 | < 0.001 | 2.06 | -0.99 | 0.323 | < 0.001 |
|            | LL        | 3.02 | -1.83 | 0.301 | < 0.001 | 2.46 | -1.31 | 0.312 | < 0.001 |
|            | SW20      | 3.70 | -2.34 | 0.166 | < 0.001 | 3.02 | -1.71 | 0.178 | < 0.001 |
|            | LL-SW20   | 2.64 | -1.54 | 0.257 | < 0.001 | 2.14 | -1.08 | 0.267 | < 0.001 |

628 Equations used to calculate the sensitivity indices: sensitivity index = a, where b is the constant of the

629 regression equation for standardized soil respirations, and soil temperature, soil water content, water

630 flux, and DOC flux: Y = aX + b, where Y is the standardized soil respiration rate, and X is standardized

632 T indicates soil temperature at a depth of 5 cm, SWC indicates soil water content, TF indicates

633 throughfall, LL indicates litter leachate, SW20 indicates soil water at a depth of 20 cm, LL- SW20

634 indicates the difference between litter leachate and soil water at a depth of 20 cm.

631 soil temperature, soil water content, water flux, or DOC flux.

Water Flux (mm)

DOC Flux (kg C ha-1)

Figure 2